# NON-REDUNDANT GRAPH NEURAL NETWORKS WITH IMPROVED EXPRESSIVENESS

## ABSTRACT

Message passing graph neural networks iteratively compute node embeddings by aggregating messages from all neighbors. This procedure can be viewed as a neural variant of the Weisfeiler-Leman method, which limits their expressive power. Moreover, oversmoothing and oversquashing restrict the number of layers these networks can effectively utilize. The repeated exchange and encoding of identical information in message passing amplifies oversquashing. We propose a novel aggregation scheme based on neighborhood trees, which allows for controlling the redundancy by pruning branches of the unfolding trees underlying standard message passing. We prove that reducing redundancy improves expressivity and experimentally show that it alleviates oversquashing. We investigate the interaction between redundancy in message passing and redundancy in computation and propose a compact representation of neighborhood trees, from which we compute node and graph embeddings via a neural tree canonization technique. Our method is provably more expressive than the Weisfeiler-Leman method, less susceptible to oversquashing than message passing neural networks, and provides high classification accuracy on widely-used benchmark datasets.

## 1 INTRODUCTION

Graph neural networks have recently emerged as the dominant technique for machine learning with graphs. The class of message passing neural networks (MPNNs) (Gilmer et al., 2017) is widely-used. It updates node embeddings layer-wise by combining the current embedding of a node with the embeddings of its neighbors involving learnable parameters. Suitable neural architectures admit a parametrization such that each layer represents an injective function encoding its input uniquely by the new embedding. In this case the MPNN has the same expressive power as the Weisfeiler-Leman algorithm (Xu et al., 2019). The Weisfeiler-Leman algorithm can distinguish two nodes if and only if the unfolding trees representing their neighborhoods are non-isomorphic. This unfolding tree corresponds to the computational tree of MPNNs (Scarselli et al., 2009; Jegelka, 2022). Hence, if two nodes have isomorphic unfolding trees of height $n$, they will obtain the same embedding after $n$ layers. Vice versa, for two nodes with non-isomorphic unfolding trees of height $n$, there are parameters of the network such that the node's embeddings after $n$ layers differ. However, in practice shallow MPNNs are widely employed. Two phenomena have been identified explaining the poor performance of deep graph neural networks. First, node representations are observed to converge to the same values for deep architecture instead of being able to distinguish more vertices, a phenomenon referred to as *oversmoothing* (Li et al., 2018; Liu et al., 2020). Second, *oversquashing* (Alon & Yahav, 2021) refers to the problem that the neighborhood of a node grows exponentially with the number of layers and aggregation steps and, therefore, cannot be supposed to be accurately represented by a fixed-sized embedding.

We argue that oversquashing can be alleviated by removing the encoding of repeated information. Consider a node $u$ with an edge $e = \{u, v\}$ in an undirected graph. In a first step, $u$ will send information to $v$ over the edge $e$. In the next step, $u$ will receive a message form $v$ via $e$ that incorporates the information that $u$ has previously sent to $v$. Clearly, this information is redundant. In the context of walk-based graph learning this problem is well-known and referred to as *tottering* (Mahé et al., 2004). Recently, Chen et al. (2022) made the relation between redundancy and oversquashing explicit and investigated it using the Jacobian of node representations (Topping et al., 2022). Several graph neural networks have been proposed replacing the walk-based aggregation with repeated

vertices by mechanisms based on simple or shortest paths (Abboud et al., 2022b; Michel et al., 2023; Jia et al., 2020). Closely related to our work are PathNNs (Michel et al., 2023) and RFGNN (Jia et al., 2020), which both define path-based trees for nodes and use custom aggregation schemes. Both approaches suffer from high computational costs and do not exploit the computational redundancy, which is a major advantage of standard MPNNs.

**Our contribution.**    We systematically investigate the information redundancy in MPNNs and develop principled techniques to avoid superfluous messages. Fundamental to our consideration is the tree representation implicitly used by MPNNs and the Weisfeiler-Leman method. First, we develop a neural tree canonization approach processing trees systematically in a bottom-up fashion and extend it to operate on directed acyclic graphs (DAGs) representing the union of multiple trees. Our approach recovers the computational graph of MPNNs for unfolding trees, but allows to avoid redundant computations in case of symmetries. Second, we apply the canonization technique to *neighborhood trees*, which are obtained from unfolding trees by deleting nodes that appear multiple times. We show that neighborhood trees allows distinguishing nodes and graphs that cannot be distinguished by the Weisfeiler-Leman method, rendering our technique more expressive than MPNNs. Our approach removes information redundancy on node level, but the non-regular structure of subtrees leads to computational challenges. Our DAG representation of neighborhood trees has size at most $O(nm)$ for input graphs with $n$ nodes and $m$ edges and allows to reuse embeddings of isomorphic subtrees to increase efficiency. Our method achieves high accuracy on several graph classification tasks.

## 2    RELATED WORK

The graph isomorphism network (GIN) (Xu et al., 2019) is an MPNN that generalizes the Weisfeiler-Leman algorithm and reaches its expressive power. The embedding of a vertex $v$ in layer $i$ of GIN is defined as

$$x_i(v) = \text{MLP}_i\left((1 + \epsilon_i) \cdot x_{i-1}(v) + \sum_{u \in N(v)} x_{i-1}(u)\right), \tag{1}$$

where the initial features $x_0(v)$ are usually acquired by applying a multi-layer perceptron (MLP) to the vertex features. The limited expressiveness of simple message passing neural networks has lead to an increased interest in researching the expressiveness of GNNs and finding more powerful architectures, for example, by encoding graph structure as additional features or altering the message passing procedure. Shortest Path Networks (Abboud et al., 2022a) use multiple aggregation functions for different shortest path lengths: One for each $k$ for the $k$-hop neighbors. While this allows the target node to directly communicate with nodes further away and in turn possibly might help mitigate oversquashing, some information about the structure of the neighborhood can still not be represented adequately and the gain in expressiveness is limited. In Distance Encoding GNNs (Li et al., 2020), the distances of the nodes to a set of target nodes are encoded. While this approach also is provably more expressive than the standard WL method, it is limited to solving node-level tasks, since the encoding depends on a fixed set of target nodes, and has not been employed for graph-level tasks. MixHop (Abu-El-Haija et al., 2019) employs an activation function for each neighborhood and concatenates their results. However, in contrast to (Abboud et al., 2022a), the aggregation is based on normalized powers of the adjacency matrix, not shortest paths, which does not solve the problem of redundant messages. SPAGAN (Yang et al., 2019) proposes a path-based attention mechanism. Although the idea is very similar, shortest paths are only sampled and the feature aggregation differs. Only one layer is used and the paths are used as features. This approach has not been investigated theoretically. Also Sun et al. (2022) uses a method with short-rooted random walks to capture long-range dependencies between nodes. It has notable limitations due to sample paths instead of exploring all of them, and the evaluation is solely on node classification datasets and needs an extensive study of their expressive power. IDGNN (You et al., 2021) keeps track of the identity of the root node in the unfolding tree, which allows for more expressiveness than 1-WL. Their variant ID-GNN-Fast works by using cycles lengths as additional node features. Both variants however, do not reduce the amount of redundant information that is aggregated over multiple layers. PathNNs (Michel et al., 2023) defines path-based trees and a custom aggregation scheme, but the computational redundancy is not exploited. In RFGNNs (Chen et al., 2022) the idea of reducing redundancy is similar. The computational flow is altered to only include each node (except for the root node) at most once in each path of the computational tree. While this reduces redundancy to some extent, nodes and even

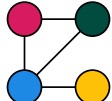 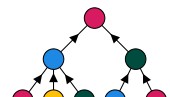 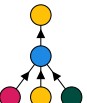 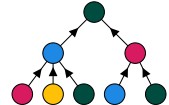 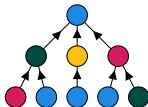

Figure 1: Graph $G$ and its unfolding trees $F_2^v$ for all $v \in V(G)$.

the same subpaths may repeatedly occur in the computational trees. The redundancy in computation is not addressed resulting in a highly inefficient preprocessing and computation, which restricts the method to a maximum of 3 layers in the experiments. See Appendix A for a detailed discussion of the differences between our approach and RFGNN.

For many of these architectures no thorough investigation on their expressiveness and connections to other approaches is provided. Moreover, these works do not explicitly investigate both types of redundancy in message passing neural networks, i.e. they do not address redundancy in the information flow and in computation.

## 3 PRELIMINARIES

In this section, we give an overview of the necessary definitions and the notation used throughout the article and introduce fundamental techniques.

**Graph theory.** A *graph* $G = (V, E, \mu, \nu)$ consists of a set of vertices $V$, a set of edges $E \subseteq V \times V$ between them and functions $\mu \colon V \to X$ and $\nu \colon E \to X$ assigning arbitrary attributes to the vertices and edges, respectively.[1] We refer to an edge from $u$ to $v$ by $uv$, and in case of undirected graphs $uv = vu$. The vertices and edges of a graph $G$ are denoted by $V(G)$ and $E(G)$, respectively, and the *neighbors* (or in-neighbors) of a vertex $u \in V$ are denoted by $N(u) = \{v \mid vu \in E\}$. The *out-neighbors* of a vertex $u \in V$ are denoted by $N_o(u) = \{v \mid uv \in E\}$. A *multigraph* is a graph, where $E$ is a multiset, which means there can be multiple edges between a pair of vertices. Two graphs $G$ and $H$ are isomorphic, denoted by $G \simeq H$, if there exists a bijection $\phi \colon V(G) \to V(H)$, so that $\forall u, v \in V(G) \colon \mu(v) = \mu(\phi(v)) \wedge uv \in E(G) \Leftrightarrow \phi(u)\phi(v) \in E(H) \wedge \forall uv \in E(G) \colon \nu(uv) = \nu(\phi(u)\phi(v))$. We call $\phi$ an *isomorphism* between $G$ and $H$.

An *in-tree* $T$ is a connected, directed, acyclic graph with a distinct vertex $r \in V(T)$ with no outgoing edges referred to as *root*, denoted by $r(T)$, in which $\forall v \in V(T)\backslash r(T) : |N_o(v)| = 1$. For $v \in V(T)\backslash r(T)$ the *parent* $p(v)$ is defined as the unique vertex $u \in N_o(v)$, and $\forall v \in V(T)$ the *children* are defined as $\mathrm{chi}(v) = N(v)$. We refer to all vertices with no incoming edges as *leaves* denoted by $l(T) = \{v \in V(T) \mid \mathrm{chi}(v) = \emptyset\}$. Conceptually it is a directed tree, in which there is a unique directed path from each vertex to the root (Mehlhorn & Sanders, 2008). In our paper, we only cover in-trees and will thereby just refer to them as *trees*. In-trees are generalized by directed, acyclic graphs (DAGs). The *leaves* of a DAG $D$ and the *children* of a vertex are defined as in trees. However, there can be multiple roots and a vertex may have more than one parent. We refer to all vertices in $D$ with no outgoing edges as *roots* denoted by $r(D) = \{v \in V(D) \mid N_o(v) = \emptyset\}$ and define the *parents* $p(v)$ of a vertex $v$ as $p(v) = N_o(v)$. The height $\mathrm{hgt}$ of a node $v$ is the length of the longest path from any leaf to $v$: $\mathrm{hgt}(v) = 0$, if $v \in l(D)$ and $\mathrm{hgt}(v) = \max_{c \in \mathrm{chi}(v)} \mathrm{hgt}(c) + 1$ otherwise. The height of a DAG $D$ is defined as $\mathrm{hgt}(D) = \max_{v \in V(D)} \mathrm{hgt}(v)$. For clarity we refer to the vertices of a DAG as nodes to distinguish them from the graphs that are the input of a graph neural network.

**Weisfeiler-Leman unfolding trees.** The 1-dimensional Weisfeiler-Leman (WL) algorithm or *color refinement* starts with all vertices having a color representing their label (or a uniform coloring in case of unlabeled vertices). In each iteration the color of a vertex is updated based on the multiset of colors of its neighbors according to

$$c_{\mathsf{wl}}^{(i+1)}(v) = h\left(c_{\mathsf{wl}}^{(i)}(v), \{\!\!\{c_{\mathsf{wl}}^{(i)}(u) \mid u \in N(v)\}\!\!\}\right) \quad \forall v \in V(G),$$

where $h$ is an injective function typically representing colors by integers.

---

[1]We do not consider edge attributes in the following for clarity of presentation. However, the methods we propose can be extended to incorporate them.

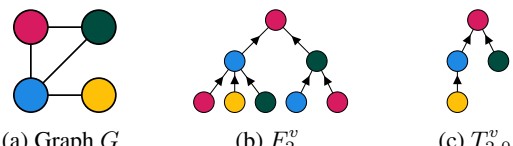

(a) Graph $G$      (b) $F_2^v$      (c) $T_{2,0}^v$      (d) $T_{2,1}^v$

Figure 2: Graph $G$ and the unfolding, 0- and 1-redundant neighborhood trees of height 2 of vertex $v$ (vertex in the upper left of $G$).

The color of a vertex of $G$ encodes its neighborhood by a tree $T$ that may contain multiple representatives of each vertex in $G$. Let $\phi\colon V(T) \to V(G)$ be a mapping such that $\phi(n) = v$ if the node $n$ in $V(T)$ represents the vertex $v$ in $V(G)$. The *unfolding tree* $F_i^v$ with height $i$ of the vertex $v \in V(G)$ consists of a root $n_v$ with $\phi(n_v) = v$ and child subtrees $F_{i-1}^u$ for all $u \in N(v)$, where $F_0^v = (\{n_v\}, \emptyset)$. The attributes of the original graph are preserved, see Figure 1 for an example. The unfolding trees $F_i^v$ and $F_i^w$ of two vertices $v$ and $w$ are isomorphic if and only if $c_{\mathsf{wl}}^{(i)}(v) = c_{\mathsf{wl}}^{(i)}(w)$.

# 4 Non-Redundant Graph Neural Networks

We propose to restrict the information flow in message passing to control redundancy using $k$-redundant neighborhood trees. We first develop a neural tree canonization technique and obtain an MPNN via its application to unfolding trees. Then we investigate computational methods on graph level, reusing information computed for subtrees and derive a customized GNN architecture. Finally, we prove that 1-redundant neighborhood trees are strictly more expressive than unfolding trees on both node- and graph-level.

## 4.1 Removing Information Redundancy

It is well-known that two nodes obtain the same WL color if and only if their unfolding trees are isomorphic and this concept directly carries over to message passing neural networks and their computational tree (Scarselli et al., 2009; Jegelka, 2022). However, unfolding trees were mainly used as a tool in expressivity analysis and as a concept explaining mathematical properties in graph learning (Kriege et al., 2016; Nikolentzos et al., 2023). We discuss a tree canonization perspective on MPNNs and derive a novel non-redundant GNN architecture based on neighborhood trees.

Aho, Hopcroft, and Ullman (1974, Section 3.2) describe a linear time isomorphism test for rooted unordered trees in their classical text book, see Appendix B for details. We give a high-level description to lay the foundations for our neural variant without focusing on the running time. The algorithm proceeds in a bottom-up fashion and assigns integers $c_{\mathsf{ahu}}(v)$ to each node $v$ of the tree. Let $f$ be a function that assigns a pair consisting of an integer and a multiset of integers injectively to a new (unused) integer. Initially, all leaves $v$ are assigned integers $c_{\mathsf{ahu}}(v) = f(\mu(v), \emptyset)$ according to their label $\mu(v)$. Then, the internal nodes are processed level-wise in a bottom-up fashion guaranteeing that whenever a node is processed all its children have been considered. Hence, the algorithm computes for all nodes $v$ of the tree

$$c_{\mathsf{ahu}}(v) = f(\mu(v), \{\!\!\{ c_{\mathsf{ahu}}(u) \mid u \in \mathrm{chi}(v) \}\!\!\}). \tag{2}$$

**GNNs via unfolding tree canonization.** We combine Eq. (2) with the definition of unfolding trees and denote the root of an unfolding tree of height $i$ of a vertex $v$ by $n_v^i$. Then, we obtain

$$c_{\mathsf{ahu}}(n_v^i) = f(\mu(n_v^i), \{\!\!\{ c_{\mathsf{ahu}}(n_u^{i-1}) \mid n_u^{i-1} \in \mathrm{chi}(n_v^i) \}\!\!\}) = f(\mu(v), \{\!\!\{ c_{\mathsf{ahu}}(n_u^{i-1}) \mid u \in N(v) \}\!\!\}). \tag{3}$$

Realizing $f$ using a suitable neural architecture and replacing its codomain by embeddings in $\mathbb{R}^d$ we immediately obtain a GNN from our canonization approach. The only notable difference to standard GNNs is that the first component of the pair in Eq. (3) is the initial node feature instead of the embedding of the previous iteration. We use the technique proposed by Xu et al. (2019) replacing the first addend in Eq. (1) with the initial embedding to obtain the *unfolding tree canonization GNN*

$$x_i(v) = \mathrm{MLP}_i \left( (1 + \epsilon_i) \cdot x_0(v) + \sum_{u \in N(v)} x_{i-1}(u) \right). \tag{4}$$

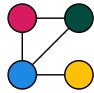 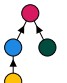 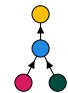 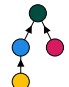 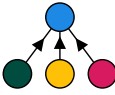

Figure 3: Graph $G$ and its 0-NTs $T_{2,0}^v$ for all $v \in V(G)$.

It is known that MPNNs cannot distinguish two nodes with the same WL color or unfolding tree. Since the function $c_{\mathsf{ahu}}(n_v^i)$ uniquely represents the unfolding tree for an injective function $f$ which can be realized by Eq. (4) (Xu et al., 2019), we conclude the following.

**Proposition 1.** *Unfolding tree canonization GNNs given by Eq. (4) are as expressive as GIN (Eq. (1)).*

However, since $x_{i-1}(v)$ represents the whole unfolding tree rooted at $v$ of height $i-1$, while using the initial node features $x_0(v)$ is sufficient, the canonization-based approach avoids redundancy. We proceed by investigating redundancy in unfolding trees themselves.

**GNNs via neighborhood tree canonization.**   We use the concept of neighborhood trees allowing to control the redundancy in unfolding trees. A $k$-redundant neighborhood tree ($k$-NT) $T_{i,k}^v$ can be constructed from the unfolding tree $F_i^v$ by deleting all subtrees with its roots, that already occurred more than $k$ levels before (seen from root to leaves). Let $\mathrm{depth}(v)$ denote the length of the path from $v$ to the root and $\phi(v)$ again denote the original vertex in $V(G)$ represented by $v$ in the unfolding or neighborhood tree.

**Definition 1** ($k$-redundant Neighborhood Tree). *For $k \geq 0$, the $k$-redundant neighborhood tree ($k$-NT) of a vertex $v \in V(G)$ with height $i$, denoted by $T_{i,k}^v$, is defined as the subtree of the unfolding tree $F_i^v$ induced by the nodes $u \in V(F_i^v)$ satisfying*

$$\forall w \in V(F_i^v)\colon \phi(u) = \phi(w) \Rightarrow \mathrm{depth}(u) \leq \mathrm{depth}(w) + k.$$

Figures 2 and 3 show a examples of unfolding and neighborhood trees. Note that for $k \geq i$ the $k$-redundant neighborhood tree is equivalent to the WL unfolding tree.

We can directly apply the neural tree canonization technique to neighborhood trees. However, a simplifying expression based on the neighbors in the input graph as given by Eq. (3) for unfolding trees is not possible for neighborhood trees. Therefore, we investigate techniques to systematically exploit computational redundancy.

## 4.2   REMOVING COMPUTATIONAL REDUNDANCY

The computation DAG of an MPNN involves the embedding of a set of trees representing the node neighborhoods of a single or multiple graphs. Results computed for one tree can be reused for others by identifying isomorphic substructures causing computational redundancy. We first describe, how to merge trees in general and then discuss the application to unfolding and neighborhood trees.

**Merging trees into a DAG.**   The neural tree canonization approach developed in the last section can directly be applied to DAGs. Given a DAG $D$, it computes an embedding for each node $n$ in $D$ that represents the tree $F_n$ obtained by recursively following its children similar as in unfolding trees, cf. Section 3. Since $D$ is acyclic the height of $F_n$ is bounded. A detailed description of a neural architecture is postponed to Section 4.3.

Given a set of trees $\mathcal{T} = \{T_1, \ldots, T_n\}$, a *merge DAG* of $\mathcal{T}$ is a pair $(D, \xi)$, where $D$ is a DAG, $\xi\colon \{1, \ldots n\} \to V(D)$ is a mapping and for all $i \in \{1, \ldots, n\}$ we have $T_i \simeq F_{\xi(i)}$. The definition guarantees that the neural tree canonization approach applied to the merge DAG produces the same result for the nodes in the DAG as for the nodes in the original trees. A trivial merge DAG is the disjoint union of the trees with $\xi(i) = r(T_i)$. However, depending on the structure of the given trees, we can identify the subtrees they have in common and represent them only once, such that two nodes of different trees share the same child, resulting in a DAG instead of a forest.

We propose an algorithm that builds a merge DAG by successively adding trees to an initially empty DAG creating new nodes only when necessary. Our approach maintains a canonical labeling for each node of the DAG and computes a canonical labeling for each node of the tree to be added using the

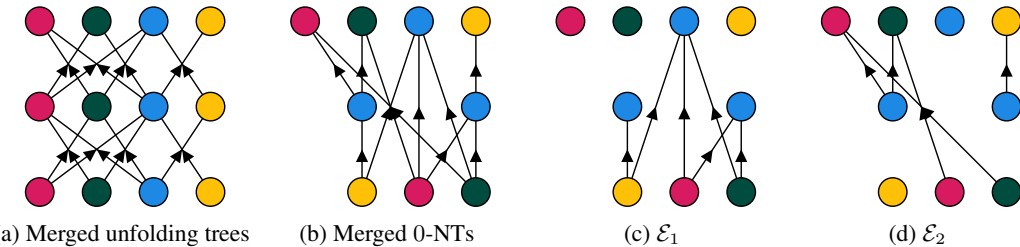

(a) Merged unfolding trees     (b) Merged 0-NTs     (c) $\mathcal{E}_1$     (d) $\mathcal{E}_2$

Figure 4: Computation DAGs for unfolding (a) and 0-NTs (b) of height 2 of graph $G$. And edges in the different layers of the merge DAG of 0-NTs (c), (d).

AHU algorithm, cf. Appendix B. Then, the tree is processed starting at the root. If the canonical labeling of the root is present in the DAG, then algorithm terminates. Otherwise the subtrees rooted at its children are inserted into the DAG by recursive calls. Finally, the root is created and connected to the representatives of its children in the DAG. We introduce a node labeling $L\colon V_T \to \mathcal{O}$ used for tree canonization, where $V_T = \bigcup_{i=1}^{n} V(T_i)$ and $\mathcal{O}$ an arbitrary set of labels, refining the original node attributes, i.e., $L(u) = L(v) \Rightarrow \mu(u) = \mu(v)$ for all $u, v$ in $V_T$. When $\mathcal{O}$ are integers from the range 1 to $|V_T|$, the algorithm runs in $O(|V_T|)$ time, see Appendix D for details. When two siblings that are the roots of isomorphic subtrees are merged, this leads to parallel edges in the DAG. Parallel edges can be avoided by using a labeling satisfying $L(u) = L(v) \Rightarrow \mu(u) = \mu(v) \land p(u) \neq p(v)$ for all $u, v$ in $V_T$.

Unfolding trees and $k$-NTs can grow exponentially in size with increasing height. However, this is not case for the merge DAGs obtained by the algorithm described above as we will show below. Moreover, we can directly generate DAGs of size $O(m \cdot (k+1))$ representing individual $k$-NTs with unbounded height in a graph with $m$ edges, see Appendix C for details.

**Merging unfolding trees.** Merging the unfolding trees of a graph with the labeling $L = \phi$ leads to the computation DAG of GNNs. Figure 4a shows the computation DAG for the graph from Figure 1. The roots in this DAG correspond to the representation of the vertices after aggregating information from the lower layers. Each node occurs once at every layer of the DAG and the links between any two consecutive layers are given by the adjacency matrix of the original graph. While this allows computation based on the adjacency matrix widely-used for MPNNs, it involves the encoding of redundant information. Our method has the potential to compress the computational DAG further by using the less restrictive labeling $L = \mu$ leading to a DAG, where at layer $i$ all vertices $u, v$ with $c_{\mathsf{wl}}^{(i)}(u) = c_{\mathsf{wl}}^{(i)}(v)$ are represented by the same node. This compression appears particularly promising for graphs with symmetries.

**Merging neighborhood trees.** Merging the $k$-redundant neighborhood trees in the same way using the labeling $L = \mu$ (or $L = \phi$ if we want to avoid parallel edges), leads to a computation DAG having a less regular structure, see Figure 4b for an example. First, there might be multiple nodes on the same level representing the same original vertex. Second, the adjacency matrix of the original graph cannot be used to propagate the information. A trivial upper bound on the size of the merge DAG of a graph with $n$ nodes and $m$ edges is $O(nmk + nm)$.

We apply the neural tree canonization approach to the merge DAG in a bottom-up fashion from the leaves to the roots. Each edge is used exactly once in this computation. Let $D = (\mathcal{V}, \mathcal{E})$ be a merge DAG. The nodes can be partitioned based on their height resulting in $\mathcal{L}_i = \{v \in \mathcal{V} \mid \mathrm{hgt}(v) = i\}$. This induces the edge partition $\mathcal{E}_i = \{uv \in \mathcal{E} \mid v \in \mathcal{L}_i\}$, in which all edges with some end node $v$ are in the same layer and all the incoming edges of children of $v$ are in a previous layer. Note, that since $\mathcal{L}_0$ contains all leaves of the DAG, there is no $\mathcal{E}_0$. Figures 4c and 4d show the edge sets $\mathcal{E}_1$ and $\mathcal{E}_2$ for the example merge DAG of Figure 4b.

### 4.3 NON-REDUNDANT NEURAL ARCHITECTURE (DAG-MLP)

We introduce a neural architecture computing embeddings for the nodes in a merge DAG allowing to retrieve embeddings of the contained trees from its roots. First, a preprocessing step transforms the

node labels using $\text{MLP}_0$, mapping them to an embedding space of fixed dimensions. Then, an $\text{MLP}_i$ is used to process the nodes at each layer $\mathcal{L}_i$.

$$\mu'(v) = \text{MLP}_0\left(\mu(v)\right), \qquad\qquad\qquad\qquad \forall v \in \mathcal{V}$$
$$x(v) = \mu'(v), \qquad\qquad\qquad\qquad\qquad \forall v \in \mathcal{L}_0$$
$$x(v) = \text{MLP}_i\left((1+\epsilon_i) \cdot \mu'(v) + \sum_{\forall u:\, uv \in \mathcal{E}_i} x(u)\right), \qquad \forall v \in \mathcal{L}_i, i \in \{1, \ldots, n\}$$

The DAG-MLP can be computed through iterated matrix-vector multiplication analogous to standard GNNs. Let $\mathbf{L}_i$ be a square matrix with ones on the diagonal at position $j$ if $v_j \in \mathcal{L}_i$, and zeros elsewhere. Let $\mathbf{E}_i$ represent the adjacency matrix of $(\mathcal{V}, \mathcal{E}_i)$, and let $\mathbf{F}$ denote the node features of $\mathcal{V}$, corresponding to the initial node labels. The transformed features $\mathbf{F}'$ are obtained using the preprocessing $\text{MLP}_0$, and $\mathbf{X}^{[i]}$ represents the updated embeddings at layer $i$ of the DAG.

$$\mathbf{F}' = \text{MLP}_0\left(\mathbf{F}\right), \quad \mathbf{X}^{[0]} = \mathbf{L}_0 \mathbf{F}',$$
$$\mathbf{X}^{[i]} = \text{MLP}_i\left((1+\epsilon_i) \cdot \mathbf{L}_i \mathbf{F}' + \mathbf{E}_i \mathbf{X}^{[i-1]}\right) + \mathbf{X}^{[i-1]},$$

In the above equation, $\text{MLP}_i$ is applied to the rows associated with nodes in $\mathcal{L}_i$. The embeddings $\mathbf{X}^{[i]}$ are initially set to zero for the inner nodes and are computed level-wise. To preserve the embeddings from all previous layers, we add $\mathbf{X}^{[i-1]}$ during the computation of $\mathbf{X}^{[i]}$. Assume the merge DAG $(D, \xi)$ contains the trees $\{T_1, \ldots, T_n\}$, then we obtain a node embedding $\mathbf{X}^{[n]}_{\xi(i)}$ for each tree $T_i$ with $i \in \{1, \ldots, n\}$. When using a single tree for each vertex of the input graph, this directly yields its final embedding. If we include trees of different height for each vertex of the input graph, we group them accordingly. We use NTs of a given fixed height (Fixed Single-Height) or all NTs of size up to a certain maximum (Combine Heights), see Appendix E for a description of the resulting architecture.

### 4.4 Expressiveness of 1-NTs

Let $\varphi$ be an isomorphism between $G$ and $H$. We call two nodes $u$ and $v$ (or edges $uw$ and $vx$) *corresponding* in an isomorphism $\varphi$, if $\varphi(u) = v$ (for edges $\varphi(u)\varphi(w) = vx$). We denote two nodes $u$ and $v$ (or edges $uw$ and $vx$) by $u \cong v$ ($uw \cong vx$, respectively), if there exists an isomorphism in which $u$ and $v$ ($uw$ and $vx$) are corresponding.

For isomorphism testing the (multi)sets of unfolding trees of two graphs (and $k$-redundant neighborhood trees, respectively) can be compared. The sets are denoted with $wl_i(G)$ and $nt_{i,k}(G)$ for iteration $i$, and defined as $wl_i(G) = \{\!\!\{F_i^v | v \in V(G)\}\!\!\}$ and $nt_{i,k}(G) = \{\!\!\{T_{i,k}^v | v \in V(G)\}\!\!\}$. If two graphs $G$ and $H$ are isomorphic, then by definition of the trees, we can find a bijection $\psi$ between their tree sets $wl_\infty(G)$ and $wl_\infty(H)$ (and $nt_{\infty,k}(H)$ and $nt_{\infty,k}(G)$, respectively), with $\forall T: T \simeq \psi(T)$, which we denote by $wl_\infty(G) = wl_\infty(H)$ ($nt_{\infty,k}(G) = nt_{\infty,k}(H)$). However, $wl_\infty(G) = wl_\infty(H) \not\Rightarrow G \simeq H$ (and also $nt_{\infty,k}(G) = nt_{\infty,k}(H) \not\Rightarrow G \simeq H$). We focus on 1-redundant neighborhood trees from now on.

**Theorem 1.** *The $1$-NT isomorphism test is more powerful than the Weisfeiler-Leman isomorphism test, i.e.,*

*1. $\forall G, H: wl_\infty(G) \neq wl_\infty(H) \Rightarrow nt_{\infty,1}(G) \neq nt_{\infty,1}(H)$*
*2. $\exists G, H: G \not\simeq H \wedge wl_\infty(G) = wl_\infty(H) \wedge nt_{\infty,1}(G) \neq nt_{\infty,1}(H)$.*

*Proof.* 1. We prove the first statement by contradiction. Assume $u \in V(G), v \in V(H)$, two nodes with $u \not\cong v$, and let $i$ be the first iteration in which $F_i^u \not\simeq F_i^v$, but $T_{i,1}^u \simeq T_{i,1}^v$. From the definition it follows that $\forall v: F_0^v \simeq T_{0,1}^v$, and also $\forall v: F_1^v \simeq T_{1,1}^v$, so $i \geq 2$.

Since $i$ is the first iteration they differed, $F_{i-1}^u \simeq F_{i-1}^v$. Any isomorphism between $F_i^u$ and $F_i^v$ can only be generated from extending an isomorphism between $F_{i-1}^u$ and $F_{i-1}^v$. Let $\varphi$ be an arbitrary isomorphism between $F_{i-1}^u$ and $F_{i-1}^v$, then, no matter how we extend it, there exists an edge in the last layer of $F_i^u$, that has no corresponding edge in the last layer of $F_i^v$ (or vice versa).

If this edge is in $T_{i,1}^u$ in the last layer, then (since $T_{i,1}^u \simeq T_{i,1}^v$) there is also a corresponding edge in $T_{i,1}^v$, which implies it is also in $F_i^v$. This would imply $F_i^u \simeq F_i^v$, contradicting the assumption.

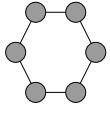  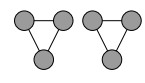  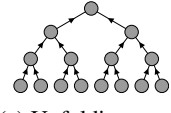  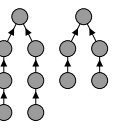

(a) Hexagon          (b) Two triangles          (c) Unfolding trees          (d) 1-NTs

Figure 5: Two graphs (a), (b) that cannot be distinguished by unfolding trees, but by 1-NTs. Figure (c) shows the unfolding tree $F_3$, which is the same for all vertices of both graphs, while (d) shows the 1-NTs of the vertices in the hexagon (left) and the triangle (right).

Table 1: Average classification accuracy for EXP-Class and CSL across $k$-folds (4-folds and 5-folds), and the number of undistinguished pairs of graphs in EXP-Iso. Best results are highlighted in gray, best results from methods with polynomial time complexity are highlighted in **bold**.

| Model | EXP-Class ↑ | EXP-Iso ↓ | CSL ↑ |
|---|---|---|---|
| GIN (Xu et al., 2019) | $50.0 \pm 0.0$ | 600 | $10.0 \pm 0.0$ |
| 3WLGNN (Maron et al., 2019) | $\mathbf{100.0 \pm 0.0}$ | **0** | $97.8 \pm 10.9$ |
| PathNN-$\mathcal{SP}^+$ (Michel et al., 2023) | $100.0 \pm 0.0$ | 0 | $100.0 \pm 0.0$ |
| PathNN-$\mathcal{AP}$ (Michel et al., 2023) | $100.0 \pm 0.0$ | 0 | $100.0 \pm 0.0$ |
| DAG-MLP (0-NTs) | $\mathbf{100.0 \pm 0.0}$ | **0** | $\mathbf{100.0 \pm 0.0}$ |
| DAG-MLP (1-NTs) | $\mathbf{100.0 \pm 0.0}$ | **0** | $\mathbf{100.0 \pm 0.0}$ |

If this edge is not in $T_{i,1}^u$ in the last layer, the same edge must have already occurred in a previous layer in $T_{i,1}^u$. Let $l$ be the layer that this edge first occurred. Then, $l \leq i-2$ must hold (because $k = 1$), and this edge must also occur in $F_i^u$, with a corresponding edge in $T_{i,1}^v$ and most importantly in $F_i^v$ in that layer (since the trees up to $i-1$ were the same). But in unfolding trees, an edge from the original graph will be present in every layer after its first occurrence. If the corresponding edge occurs in $F_i^v$ in layer $l$, it also has to occur in layer $i$ again (implying $F_i^u \simeq F_i^v$), which implies $T_{i,1}^u \not\simeq T_{i,1}^v$ and thereby contradicts the initial assumption. So $\forall G, H : wl_\infty(G) \neq wl_\infty(H) \Rightarrow nt_{\infty,1}(G) \neq nt_{\infty,1}(H)$.

2. The statement is proven by the example, where $G$ is a hexagon and $H$ consists of two triangles. For these graphs all nodes have isomorphic unfolding trees, while their 1-NTs differ (see Figure 5).   □

The 1-NTs can also distinguish the molecules decalin and bicyclopentyl, which WL cannot distinguish. We investigate the expressiveness of 0-NTs in Appendix G.

## 5 EXPERIMENTAL EVALUATION

We evaluate DAG-MLP with $k$-NTs on both synthetic (Abboud et al., 2021; Murphy et al., 2019) and real-world datasets (Morris et al., 2020). We provide information on the datasets in Appendix H.

**Experimental setup.** For synthetic datasets, we choose the number of layers in DAG-MLP based on the average graph diameter. This ensures that most nodes can effectively aggregate information from all other nodes during message propagation. The embeddings at each layer are extracted using readouts, concatenated, and then passed to two learnable linear layers for prediction. For evaluation on the TUDataset, we follow the same splits across 10 folds as proposed by Errica et al. (2020). This allows conducting a grid search to identify the optimal hyper-parameters for each dataset. The architecture for combined heights is designed such that each "readout$_i$" is used to extract the embeddings corresponding to each layer, and the mean of the average-pooled embeddings is passed to a final MLP layer responsible for prediction (see Appendix E). For the fixed single-height architecture, only the last readout is used, pooled and passed to the final MLP layer. The hyper-parameters are detailed in Appendix J.

**Results.** Table 1 shows the results on the synthetic expressivity datasets. Comparing our approach to GIN, the results we see are consistent with the theoretical findings: 1-NTs are more expressive than GIN. Our hypothesis that 0-NTs are more expressive than GIN on a graph-level is experimentally validated, but a theoretical proof remains future work.

Table 2: Average accuracy for DAG-MLP using 4-fold cross-validation on EXP-Class (Abboud et al., 2021), evaluated with varying number of layers.

| $k$-NTs | 1 layer | 2 layers | 3 layers | 4 layers | 5 layers | 6 layers |
|---------|---------|----------|----------|----------|----------|----------|
| 0-NTs | $51.1 \pm 1.6$ | $57.5 \pm 6.6$ | $91.7 \pm 11.6$ | $99.7 \pm 0.3$ | $\mathbf{100.0 \pm 0.0}$ | $\mathbf{100.0 \pm 0.0}$ |
| 1-NTs | $50.1 \pm 0.2$ | $58.9 \pm 4.6$ | $59.4 \pm 5.7$ | $99.6 \pm 0.5$ | $99.9 \pm 0.2$ | $\mathbf{100.0 \pm 0.0}$ |
| 2-NTs | - | $52.6 \pm 3.4$ | $54.9 \pm 5.3$ | $52.4 \pm 3.8$ | $97.6 \pm 1.9$ | $\mathbf{100.0 \pm 0.0}$ |
| 3-NTs | - | - | $56.2 \pm 5.7$ | $51.1 \pm 1.9$ | $52.4 \pm 4.1$ | $87.1 \pm 21.4$ |
| 4-NTs | - | - | - | $50.1 \pm 0.2$ | $50.6 \pm 1.0$ | $50.4 \pm 0.7$ |
| 5-NTs | - | - | - | - | $50.4 \pm 0.7$ | $50.0 \pm 0.0$ |
| 6-NTs | - | - | - | - | - | $53.2 \pm 5.2$ |

Table 3: Classification accuracy ($\pm$ standard deviation) over 10-fold cross-validation on the datasets from TUDataset, taken from Michel et al. (2023). Best performance is highlighted in gray, best results from methods with polynomial time complexity are highlighted in **bold**. "-" denotes not applicable and "NA" means not available.

| | Algorithm | IMDB-B | IMDB-M | ENZYMES | PROTEINS |
|---|-----------|--------|--------|---------|----------|
| Linear | GIN (Xu et al., 2019) | $71.2 \pm 3.9$ | $48.5 \pm 3.3$ | $59.6 \pm 4.5$ | $73.3 \pm 4.0$ |
| | GAT (Veličković et al., 2018) | $69.2 \pm 4.8$ | $48.2 \pm 4.9$ | $49.5 \pm 8.9$ | $70.9 \pm 2.7$ |
| | SPN ($l = 1$) (Abboud et al., 2022b) | NA | NA | $67.5 \pm 5.5$ | $71.0 \pm 3.7$ |
| | SPN ($l = 5$) (Abboud et al., 2022b) | NA | NA | $69.4 \pm 6.2$ | $\mathbf{74.2 \pm 2.7}$ |
| Exp | PathNet (Sun et al., 2022) | $70.4 \pm 3.8$ | $49.1 \pm 3.6$ | $69.3 \pm 5.4$ | $70.5 \pm 3.9$ |
| | PathNN-$\mathcal{P}$ (Michel et al., 2023) | $72.6 \pm 3.3$ | $50.8 \pm 4.5$ | $73.0 \pm 5.2$ | $75.2 \pm 3.9$ |
| | PathNN-$\mathcal{SP}^+$ (Michel et al., 2023) | - | - | $70.4 \pm 3.1$ | $73.2 \pm 3.3$ |
| Ours | DAG-MLP (0-NTs) Fixed Single-Height | $\mathbf{72.9 \pm 5.0}$ | $50.2 \pm 3.2$ | $67.9 \pm 5.3$ | $70.1 \pm 1.7$ |
| | DAG-MLP (1-NTs) Fixed Single-Height | $72.4 \pm 3.8$ | $48.8 \pm 4.3$ | $\mathbf{70.6 \pm 5.5}$ | $70.2 \pm 3.4$ |
| | DAG-MLP (0-NTs) Combine Heights | $72.8 \pm 5.6$ | $50.1 \pm 3.8$ | $66.7 \pm 4.8$ | $69.1 \pm 3.6$ |
| | DAG-MLP (1-NTs) Combine Heights | $72.2 \pm 4.5$ | $\mathbf{51.3 \pm 4.4}$ | $69.2 \pm 4.5$ | $69.5 \pm 3.0$ |

In Table 2, we investigate the impact of the parameter $k$ and the number of layers $l$ on the accuracy on EXP-Class. Cases with $k > l$ can be disregarded, since the computation for NTs remains the same when $k = l$. Empirically, 0- and 1-NTs yield the highest accuracy. This is consistent with our discussions on expressivity in Section 4.4 and Appendix G. The decrease in accuracy with increasing $k$ indicates that information redundancy leads to oversquashing.

For TUDataset, we report the accuracy compared to related work in Table 3. Due to the high standard deviation across all methods, we present a statistical box plot for the accuracy based on three runs on the testing set of 10-fold cross-validation in Appendix F. We group the methods by their time complexity. Note that, while PathNN performs well on ENZYMES and PROTEINS, the time complexity of this method is exponential. Therefore, we also highlight the best method with polynomial time complexity. For IMDB-B and IMDB-M, which have small diameters, we see that $k$-NTs outperform all other methods. For ENZYMES a variant of our approach achieves the best result among the approaches with non-exponential time complexity and $k$-NTs lead to a significant improvement over GIN.

## 6  CONCLUSION

We propose a neural tree canonization technique and combined it with neighborhood trees, which are pruned and more expressive versions of unfolding trees used by standard MPNNs. By merging trees in a DAG, we derive compact representations that form the basis for our neural architecture DAG-MLP, which learns across DAG levels. It inherits the properties of the GIN architecture, but is provably more expressive than 1-WL when based on 1-redundant neighborhood trees. In this effort, we introduced general techniques to derive compact computation DAGs for tree structures encoding node neighborhoods. This revealed a complex interplay between information redundancy, computational redundancy and expressivity, the balancing of which is an avenue for future work.

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

## A    COMPARISON TO RFGNN AND TPTS

In RFGNN (Chen et al., 2022), so called TPTs are used to represent the information flow. The motivation of the approach is similar to ours, but there are substantial differences between TPTs and $k$-redundant neighborhood trees and the computational properties of the techniques. In RFGNN the focus is solely on reducing redundancy in information flow, not computation. Additionally, the definition of TPTs allows for much more redundancy than that of $k$-NTs. We first introduce the concepts used by Chen et al. (2022), and then discuss differences and disadvantages.

An *epath* is a path with no repeated vertices, except the starting node is allowed to be the ending node if the length of the epath is larger than 2.

**Definition 2** (Truncated ePath Tree (Chen et al., 2022)). *Given graph $G$ and $v \in V(G)$, the TPT $TPT_{G,v}^h$ with height $h$ is an epath search tree obtained by running a BFS from $v$, where all epaths of length up to $h$ are accessed.*

First of all, the definition of TPTs allows for vertices to appear (redundantly) multiple times in a tree. If a vertex appeared at depth 1, for example, it can still appear anywhere else in the TPT, but not as its own descendant. Parts of paths that differ, will be repeated in TPTs, without the ability to compress them. Neighborhood trees, in contrast, allow compressed representations, see Appendix C. Chen et al. (2022, Lemma 7) show that TPTs are more expressive than unfolding trees of the same height by giving two example graphs, which can also be distinguished by the more 1-NTs of the same height, cf. Theorem 1.

The size and running time complexity of RFGNN is very restrictive. While the term BFS in the definition suggests a linear running time, the BFS has to be modified leading to an exponential running time, cf. Appendix C. Compression of the TPTs (or even the forest of TPTs from the vertices of a graph) is not discussed in the paper, making preprocessing, as well as computation much more time consuming. In the experimental evaluation, TPTs only up to height 3 are used due to the resource-intensity, which shows that the full expressivity of TPTs cannot be utilized in practice. This is also seen in the experimental evaluation in (Chen et al., 2022), where the results using RFGNN are only marginally better.

With our proposed approach, we investigate not only the redundancy in information flow using the $k$-NTs, but also remove redundancy in computation by employing merge DAGs. Our approach is capable of reaching its full expressive potential in practice with a reasonable running time.

## B    THE AHU ALGORITHM

Aho, Hopcroft and Ullman describe a linear-time algorithm for deciding whether two rooted unordered trees are isomorphic (Aho et al., 1974, Section 3.2). The concept of the algorithm inspired our neural tree canonization algorithm, see Section 4.1, and the algorithm provides a fundamental subroutine of our method for combining trees into one merge DAG, see Appendix D. Here, we give a complete description of the original algorithm, its extension to trees with node labels or features and the required modification for tree canonization.

In its original version, the algorithm solves the subtree isomorphism problem for two rooted unordered unlabeled trees $T_1$ and $T_2$. Algorithm 1 shows the pseudocode of the algorithm.[2] First, the nodes in the disjoint union of the input trees $T_1 \cup T_2$ are partitioned into levels according to their depth distinguishing leaves and internal nodes, see Figure 6. Note that the levels are numbered in reverse order of depth, i.e., for a node $v$ on level $i$ the equality $\mathrm{depth}(v) = \mathrm{hgt}(T) - i$ holds. The lists $\mathcal{L}_i^*$ and $\mathcal{L}_i$ contain all leaves and internal nodes, respectively, on level $i$. The labels $\mathsf{c}_{\mathsf{ahu}}$ of the leaves are set to 0 and the tree is processed in bottom-up-fashion. In iteration $i$ of the for loop, the labels of all nodes $\mathcal{L}_k$ for all $k < i$ have been computed and $\mathcal{L}_k$ is sorted according to them. Note that $\mathcal{L}_i^*$ contains only nodes $v$ with $\mathsf{c}_{\mathsf{ahu}}(v) = 0$ for all $0 \le i \le \mathrm{hgt}(T)$. Tuples $\hat{\mathsf{c}}_{\mathsf{ahu}}(v)$ are generated for the nodes $v$ in $\mathcal{L}_i$ by iterating over $\mathcal{L}_{i-1}^*$ and then $\mathcal{L}_{i-1}$ appending the label of the current node to the tuple of its parent. Each tuple contains an integer label for each child and is in non-decreasing

---

[2]For clarity of presentation, we adapted and simplified the textual description of the textbook (Aho et al., 1974). In contrast to the original description, our algorithm operates on the disjoint union of both trees instead of applying the same operations to $T_1$ and $T_2$ individually.

---

**Algorithm 1** AHU algorithm for tree isomorphism

---

**function** RELABEL(nodes $\mathcal{L}$, labels $\hat{c}_{\mathsf{ahu}}$, $c_{\mathsf{ahu}}$)   ▷ Replaces tuple labels $\hat{c}_{\mathsf{ahu}}(v)$ by integer labels $c_{\mathsf{ahu}}(v)$ for the nodes $v$ in the list $\mathcal{L} = (l_1, l_2, \ldots, l_N)$ sorted according to $\hat{c}_{\mathsf{ahu}}$.
    prev $\leftarrow \hat{c}_{\mathsf{ahu}}(l_1)$
    $k \leftarrow 1$
    **for** $i \leftarrow 1$ **to** $N$ **do**
        **if** $\hat{c}_{\mathsf{ahu}}(l_i) = $ prev **then**
            $c_{\mathsf{ahu}}(l_i) \leftarrow k$
        **else**
            $k \leftarrow k + 1$                            ▷ Next integer label
            $c_{\mathsf{ahu}}(l_i) \leftarrow k$
        prev $\leftarrow \hat{c}_{\mathsf{ahu}}(l_i)$
    **return** $c_{\mathsf{ahu}}$

**function** TREEISMORPHISM($T_1, T_2$)
    $T \leftarrow T_1 \cup T_2$
    $H \leftarrow \mathrm{hgt}(T)$
    **for** $i \leftarrow 0$ **to** $H$ **do**
        $\mathcal{L}_i^* \leftarrow \{v \in V(T) \mid \mathrm{depth}(v) = H - i \wedge \mathrm{chi}(v) = \emptyset\}$     ▷ Leaves with the same depth
        $\mathcal{L}_i \leftarrow \{v \in V(T) \mid \mathrm{depth}(v) = H - i \wedge \mathrm{chi}(v) \neq \emptyset\}$  ▷ Non-leaves with the same depth
    **for each** leaf $v \in V(T)$ **do**
        $c_{\mathsf{ahu}}(v) \leftarrow 0$                                 ▷ Initialize labels for leaves
    **for** $i \leftarrow 1$ **to** $H$ **do**
        **for each** $l$ in ordered list $\mathcal{L}_{i-1}^* + \mathcal{L}_{i-1}$ **do**        ▷ Iterate over concatenated ordered list
            Append $c_{\mathsf{ahu}}(l)$ to the tuple $\hat{c}_{\mathsf{ahu}}(p(l))$        ▷ Pass label of previous level upwards
        $\mathcal{L}_i \leftarrow$ RADIXSORT($\mathcal{L}_i$, $\hat{c}_{\mathsf{ahu}}$)                   ▷ Sort list according to their tuples
        $c_{\mathsf{ahu}} \leftarrow$ RELABEL($\mathcal{L}_i$, $c_{\mathsf{ahu}}$, $\hat{c}_{\mathsf{ahu}}$)
        **if** $\{\!\{c_{\mathsf{ahu}}(v) \mid (\mathcal{L}_i^* \cup \mathcal{L}_i) \cap V(T_1)\}\!\} \neq \{\!\{c_{\mathsf{ahu}}(v) \mid (\mathcal{L}_i^* \cup \mathcal{L}_i) \cap V(T_2)\}\!\}$ **then**
            **return** `false`
    **return** `true`

---

order. In the next step, the nodes in $\mathcal{L}_i$ are sorted according to the tuples using radix sort. Then, the RELABEL function assigns new integers $c_{\mathsf{ahu}}(v)$ to all nodes $v$ in $\mathcal{L}_i$ based on their tuples. Since $\mathcal{L}_i$ is sorted, all nodes with the same label form a contiguous sub-list. New integers are computed by scanning the list assigning 1 to the first entry and increasing the integer whenever the current tuple differs from the previous one. Using this approach, the RELABEL function computes an injection between tuples and integers appearing for the nodes in $\mathcal{L}_i$. Two trees are isomorphic if and only if their nodes yield the same multiset of labels on all levels. Figure 6 shows an example of two trees that are identified as isomorphic. The algorithm can be implemented in linear time applying RADIXSORT to sort tuples of labels from a bounded range.

As noted by Aho et al. (1974), the algorithm can be extended to trees with initial integer labels from the range 1 to $n$ with $n = O(|V(T)|)$ by including the label of a node as the first element in its tuple. In this case, the RELABEL function assigns integers that were not used as initial labels and the leaves in $\mathcal{L}_i^*$ have to be sorted according to their label after initialization. The overall running time remains linear.[3] If the labels are not integers from a bounded range, e.g., continuous values, an initial mapping to integers is required, which can be realized by comparison-based sorting in $O(n \log n)$.

In order to generalize the method to tree canonization, it is no longer sufficient that the relabeling function is injective for the tuples appearing on each level, but it has to be injective for all possible tuples that can occur in *any* tree. We discuss this situation in Section 4.1 and propose a learnable function with this property.

---

[3]A similar technique including level-wise processing, creation of tuples sorted by radix sort and relabeling has been proposed by Shervashidze et al. (2011, Section 2.1) in the context of the Weisfeiler-Leman kernel to achieve a linear running time.

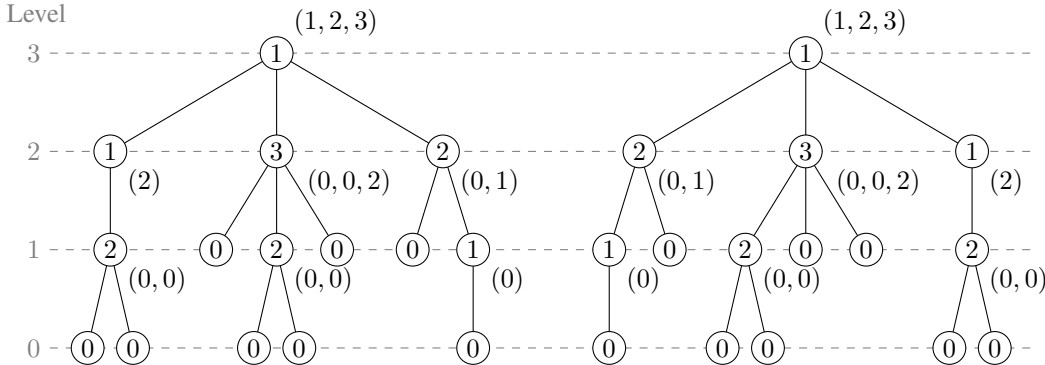

Figure 6: Two isomorphic trees $T_1$ (left) and $T_2$ (right) and the labels $c_{ahu}$ (inside each node) and $\hat{c}_{ahu}$ (right of each node) computed by the AHU algorithm.

---

**Algorithm 2** Merging trees

---

    **function** MERGE(set of trees $\mathcal{T}$, labeling $L$)                    $\triangleright$ merges $\mathcal{T}$ into a DAG $D$

         $D \leftarrow$ empty DAG                    $\triangleright$ start with empty DAG

         initialize $D.can\_map$ as an empty map          $\triangleright$ maps canonization to node in DAG

         **for each** $T \in \mathcal{T}$ **do**

             compute canonization $can(v)$ for $v \in V(T)$ under $L$

             ADD($D, T, r(T), L$)               $\triangleright$ add tree, starting at root

         **return** $D$

 

    **function** ADD(DAG $D$, tree $T$, vertex $v$, labeling $L$) $\triangleright$ adds substructure rooted at $v \in V(T)$ to $D$

         **if** $can(v) \in D.can\_map$ **then**          $\triangleright$ node (and substructure) already present in $D$

             **return**

         **for each** $c \in \text{chi}(v)$ **do**              $\triangleright$ add all children first (if necessary)

             ADD($D, T, c, L$)

         add new node $v_2$ with $L(v_2) = L(v)$ to $D$          $\triangleright$ add new node

         set $can(v_2) = can(v)$ and $D.can\_map(can(v_2)) = v_2$

         **for each** $c \in \text{chi}(v)$ **do**

             **if** edge exists from $D.can\_map(can(c))$ to $v_2$ **then**

                 increase multiplicity of that edge by 1          $\triangleright$ a sibling had the same canonization

             **else**

                 add edge from $D.can\_map(can(c))$ to $v_2$      $\triangleright$ add edges from children to new node

---

## C    BUILDING COMPACT TREES

Since unfolding trees can grow exponentially in size and our goal is to avoid redundant computation, we do not build unfolding trees and $k$-NTs explicitly. Rather, we build DAGs that represent them, corresponding to the merge DAG of only that tree using $L = \phi$. This way, the size of $k$-NTs is in $O(|E(G)| \cdot (k+1))$, which means it is linear in the size of the input graph $G$.

## D    MERGING TREES – ALGORITHM

Algorithm 2 describes how to merge a set of trees $\{T_1, \ldots, T_n\}$ into a DAG under a labeling function $L \colon \bigcup_{i \in \{1,\ldots,n\}} V(T_i) \to \mathcal{O}$, where $\mathcal{O}$ is some arbitrary labeling. All substructures that are isomorphic under $L$ are merged. For that, the canonization of all vertices is computed first. Then each tree is merged to the DAG separately: Starting at the root $r(T)$ of the tree that is added, if a node with the same canonization as $r(T)$ exists in the DAG, we do not need to do anything. Otherwise, we add the subtrees rooted at the children of $r(T)$ first (using the same procedure as for $r(T)$), and then add a new node for $r(T)$ along with edges to the nodes in the DAG that have the same canonization as the children of $r(T)$. Note that, if some children have the same canonization, in this step multiedges

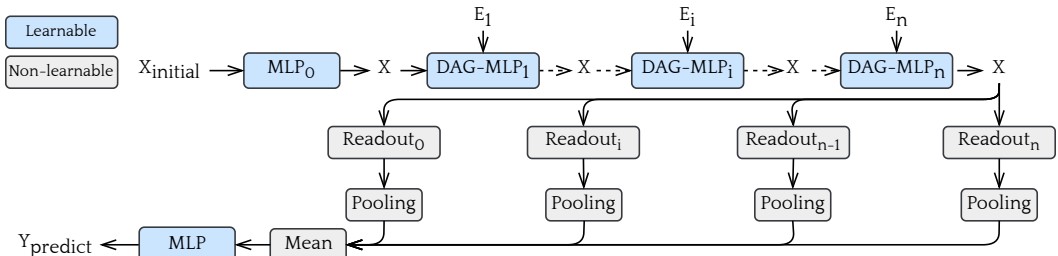

Figure 7: DAG-MLP architecture with $n$ layers for graph-level prediction tasks.

Table 4: Average accuracy of DAG-MLP using 5-fold cross-validation on CSL (Murphy et al., 2019), evaluated with varying parameters $k$ and $l$.

| $k$-NTs | 1 layer | 2 layers | 3 layers | 4 layers | 5 layers | 6 layers |
|---|---|---|---|---|---|---|
| 0-NTs | $10.0 \pm 0.0$ | $20.0 \pm 0.0$ | $40.0 \pm 0.0$ | $70.0 \pm 0.0$ | $84.0 \pm 4.9$ | $\mathbf{100.0} \pm 0.0$ |
| 1-NTs | $10.0 \pm 0.0$ | $10.0 \pm 0.0$ | $30.0 \pm 0.0$ | $48.0 \pm 4.0$ | $78.0 \pm 9.8$ | $\mathbf{100.0} \pm 0.0$ |
| 2-NTs | - | $10.0 \pm 0.0$ | $16.0 \pm 4.9$ | $20.0 \pm 12.6$ | $50.0 \pm 8.9$ | $80.0 \pm 12.6$ |
| 3-NTs | - | - | $10.0 \pm 0.0$ | $10.0 \pm 0.0$ | $20.0 \pm 12.6$ | $38.0 \pm 14.7$ |
| 4-NTs | - | - | - | $10.0 \pm 0.0$ | $16.0 \pm 8.0$ | $34.0 \pm 12.0$ |
| 5-NTs | - | - | - | - | $10.0 \pm 0.0$ | $10.0 \pm 0.0$ |
| 6-NTs | - | - | - | - | - | $10.0 \pm 0.0$ |

can occur. The algorithm can easily be extended to merge DAGs by iterating over all roots in MERGE and adding them to the DAG. The running time of the algorithm depends on the canonization, which can be done in time linear in the numbers of vertices (see Appendix B), and the time needed to add the trees to the DAG. Since we add each vertex at most once, and can check whether a canonization is already present in the DAG in constant time, this also only needs time linear in the number of tree vertices.

## E    DAG-MLP ARCHITECTURE FOR GRAPH CLASSIFICATION

Figure 7 shows an example of the architecture when using unfolding or neighborhood trees of height up to $n$ for graph classification requiring $n$ DAG-MLP layers. The node features are initially transformed into embedding with fixed dimension using $MLP_0$. Messages are then propagated using the DAG from height 0 to 1 ($\mathcal{E}_1$), which corresponds to layer 1. This process is repeated for $n$ layers, where the $i$th step computes embeddings for nodes of height $i$ in the DAG. After $n$ layers, all node embeddings in the DAG ($X$) have been updated. Using readouts, we extract the embeddings of each node from $k$-NTs of different heights within the DAG. These extracted embeddings correspond to the embeddings of different layers. A pooling operation is then applied to the output of each layer, and the pooled outputs are averaged. These averaged outputs are passed through a final MLP, transforming them into probabilities for class prediction.

## F    ADDITIONAL EXPERIMENTS

Following the same experimental setup as in Table 2, Table 4 shows the accuracy with varying parameters $k$ and $l$. Since the expressive capabilities are the same as those of GIN when the number of layers $l$ equals the redundancy parameter $k$, all results with $l = k$ are not better than guessing.

Table 5 shows a comparison of 0- or 1-NTs, with combined heights and fixed single-height, for 10-fold cross-validation on MUTAG. The results as well as those shown in Table 3 indicate that using multiple different tree heights does not improve the generalization capabilities of the model.

Figure 8 shows box plot charts for the accuracy obtained in Table 3. Due to the use of 10-fold cross-validation and the random initialization of the MLPs, the results tend to have high variance.

Table 5: Classification accuracy for 10-folds ($\pm$ standard deviation) on MUTAG comparing DAG-MLP that combines layers of different heights to DAG-MLP that only uses layers at a fixed height.

| $k$-NTs | Combine Heights | | | Fixed Single-Height | | |
|---|---|---|---|---|---|---|
| | 1 layer | 2 layers | 3 layers | 1 layer | 2 layers | 3 layers |
| 0-NTs | $84.6 \pm 6.2$ | $86.7 \pm 5.3$ | $86.9 \pm 6.0$ | $85.3 \pm 6.3$ | $89.0 \pm 4.7$ | $87.2 \pm 5.1$ |
| 1-NTs | $84.9 \pm 6.0$ | $83.3 \pm 7.3$ | $88.6 \pm 6.7$ | $85.8 \pm 6.0$ | $88.8 \pm 4.4$ | $90.4 \pm 5.1$ |

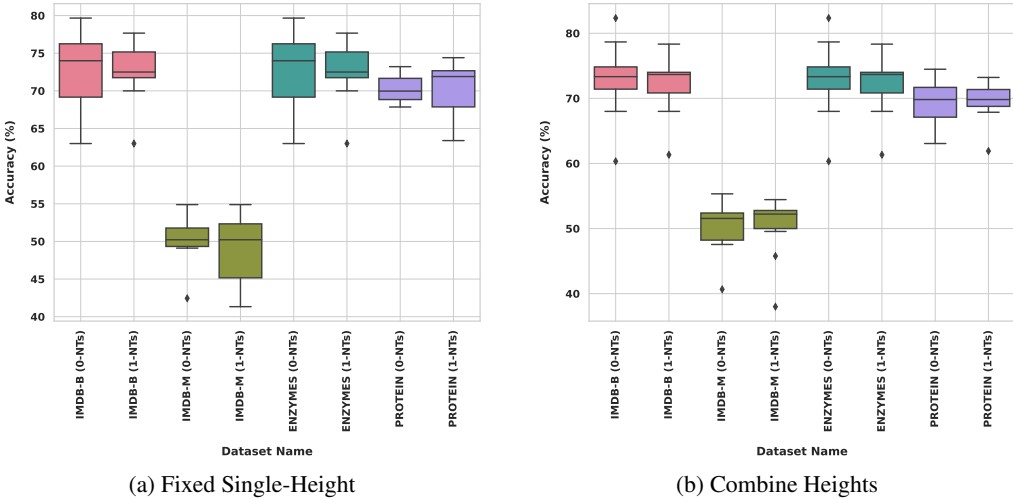

(a) Fixed Single-Height

(b) Combine Heights

Figure 8: Graph classification test accuracy box plot over three runs of 10-fold cross-validation of the DAG-MLP on the datasets from the TUDataset.

For all datasets, the accuracy of DAG-MLP is statistically within the same boundaries as those of the best related methods reported in Table 3.

## G    EXPRESSIVENESS OF 0-NTs AS NODE INVARIANTS

Theorem 1 shows that 1-NTs are more expressive as a node invariant (and in turn graph invariant) than unfolding trees. While the 0-NTs can also distinguish the nodes of the two graphs in the example of Figure 5 (and the famous example of decalin and bicyclopentyl), as a node invariant, they are not more expressive than unfolding trees in every case. Figure 9 shows an example, where the unfolding trees of two (not corresponding) vertices differ, but their 0-NTs do not. Looking at the tree sets, however, $nt_{\infty,0}$ can also distinguish those two graphs. It remains future work to delineate the expressivity of $nt_{\infty,0}$ and $wl_\infty$.

## H    DATASETS

We provide information about the datasets used in the experimental evaluation. Table 6 provides an overview of the datasets, along with their corresponding characteristics.

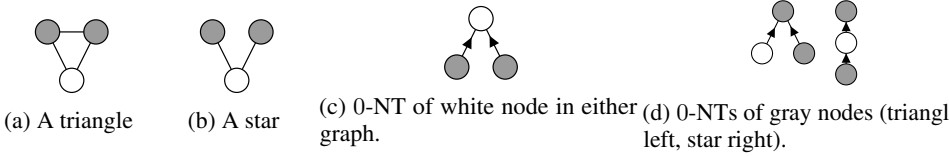

(a) A triangle          (b) A star          (c) 0-NT of white node in either graph.          (d) 0-NTs of gray nodes (triangle left, star right).

Figure 9: Graphs with vertices, not distinguishable by 0-NTs, but by unfolding trees.

Table 6: Summary of characteristics for the synthetic datasets (Murphy et al., 2019; Abboud et al., 2021) and TUDatasets (Morris et al., 2020). The table provides information on the dataset name, number of graphs ($|\mathbf{G}|$), average number of nodes ($\overline{|\mathbf{V}|}$), average number of edges ($\overline{|\mathbf{E}|}$), and average diameter ($\overline{\mathbf{D}}$) for each dataset.

| Dataset | $|\mathbf{G}|$ | $\overline{|\mathbf{V}|}$ | $\overline{|\mathbf{E}|}$ | $\overline{\mathbf{D}}$ |
|---|---|---|---|---|
| **CSL** | 150 | 41.0 | 164.0 | 6.0 |
| **EXP-Class** | 1200 | 55.8 | 139.6 | 12.6 |
| **EXP-Iso** | 1200 | 44.4 | 110.2 | 8.5 |
| **MUTAG** | 188 | 17.93 | 39.59 | 8.22 |
| **IMDB-B** | 1000 | 19.8 | 193.1 | 1.9 |
| **IMDB-M** | 1500 | 13.0 | 131.9 | 1.5 |
| **ENZYMES** | 600 | 32.6 | 124.3 | 10.9 |
| **PROTEINS** | 1113 | 39.1 | 145.6 | 11.6 |

**Synthetic datasets.** (1) EXP-Classification (EXP-Class) and EXP-Isomorphic (EXP-Iso) evaluate GNN expressivity, featuring graph pairs with varying SAT outcomes and 1-WL distinguishability (Abboud et al., 2021). EXP-Class extends EXP-Iso by including 50% "corrupted" data, making the learning task more challenging. (2) Circulant Skip Links (CSL) graphs (Murphy et al., 2019) are highly symmetric, 4-regular graphs that consist of a cycle with additional 'skip links.' Despite their symmetry, these graphs present a challenge for the WL test and GNNs based on WL, as these methods fail to distinguish between non-isomorphic instances of such graphs.

**Real-world datasets.** We examine MUTAG, IMDB-B, IMDB-M, ENZYMES, and PROTEINS from TUDataset (Morris et al., 2020). IMDB-B and IMDB-M are movie network datasets for binary and multi-class classification, respectively. ENZYMES has six protein graph classes, while PROTEINS represents a binary classification task from bioinformatics.

## I RUNNING TIME

The running time for generating and merging 0- and 1-NTs with different layers on different datasets is presented in Figure 10. We employ a parallelized algorithm to construct the NTs, where each graph is also processed in parallel.

## J HYPER-PARAMETERS

The hyper-parameters used for the synthetic datasets can be seen in Table 7. The hyper-parameters for the TUDataset experiments were chosen as follows: The batch size for training is set to $64$. Learning rate (LR) is set to $0.001$. The classifier trained for $500$ epochs. The dimension of the embedding is set to $128$. The optimizer used is Adam, and the scheduler is set to `StepLR` with a step size of $100$ and a gamma value of $0.5$. The aggregation method (pooling) is defined as `mean` and a dropout rate of $0.5$ is specified. Early stopping is configured with a patience of $250$ epochs and uses accuracy instead of loss. The number of layers for each dataset is set as in Table 8, and shuffling of the dataset is enabled.

Table 7: Synthetic dataset hyper-parameter configuration details.

| Dataset | Task | Embedding | Target | Layers | Batch Size | Epochs | LR |
|---|---|---|---|---|---|---|---|
| **EXP-Class** | Classification | 64 | 10 | 15 | 32 | 200 | $10^{-3}$ |
| **EXP-Iso** | Isomorphism Test | 1 | 1 | 6 | 1 | - | - |
| **CSL** | Classification | 64 | 10 | 6 | 32 | 200 | $10^{-3}$ |

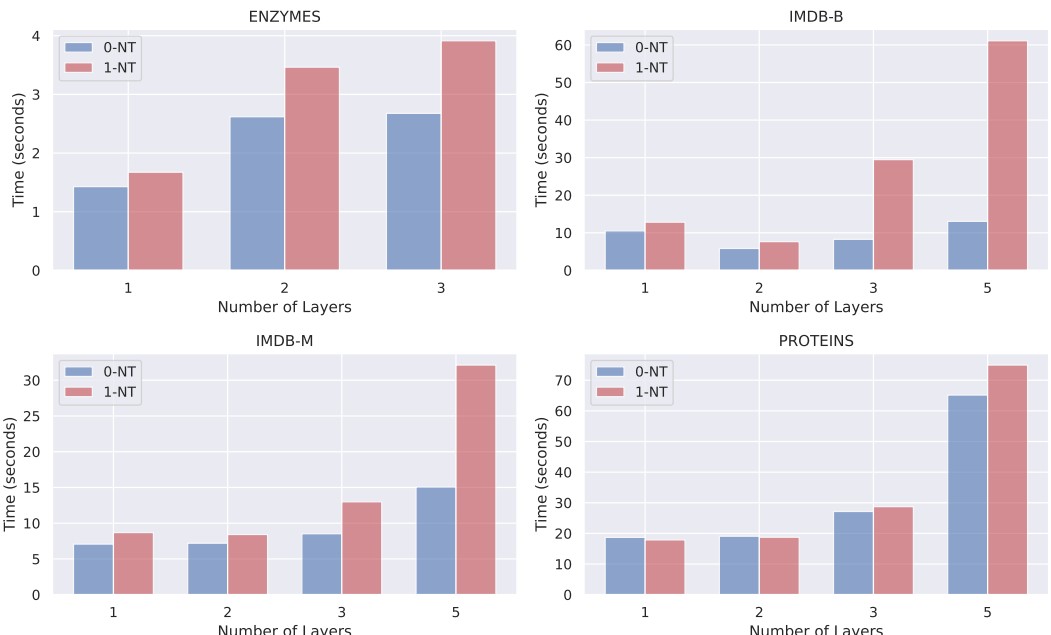

Figure 10: Running time for building the $0$- and $1$-redundant NTs.

Table 8: TUDatasets layer configuration details.

| Dataset | IMDB-B | IMDB-M | ENZYMES | PROTEINS |
|---------|--------|--------|---------|----------|
| **Layers** | 5, 3, 2 | 5, 3, 2 | 3, 2 | 5, 3, 2 |

## K  HARDWARE

The hardware configuration consists of dual AMD 7252 CPUs, each with 8 cores, and two NVIDIA A40 GPUs. The system is supplemented with 256 GB of RAM. Each NVIDIA A40 GPU comes with 10,752 CUDA cores and a clock frequency of 1.305 GHz. The GPUs have 48 GB of memory and a bandwidth of 696 GB/s, operating at a Thermal Design Power (TDP) of 300 Watts. In terms of performance, the GPUs can deliver 37,400 GFLOPs in single-precision (FP32) and 1,169 GFLOPs in double-precision (FP64) computations.

