# OpenReview forum: "Non-Redundant Graph Neural Networks with Improved Expressiveness"
_ICLR.cc/2024/Conference — Submitted to ICLR 2024_

### Official Review · Reviewer_h7VD · 2023-10-26

**Soundness:** 2 fair
**Presentation:** 2 fair
**Contribution:** 2 fair
**Rating:** 3
**Confidence:** 4

**Summary:**

This paper points that the redundancy, i.e., repeated exchange and encoding of identical information, in the message passing framework amplifies the over-squashing. To resolve the redundancy, the authors propose an aggregation scheme based on `neighborhood trees', which control redundancy by pruning branches. Authors have theoretically proved that reducing redundancy improves the expressivity, and experimentally showed it can alleviate over-squashing.

**Strengths:**

1. The paper has pointed out the inherent problem of message passing problem, the “repeated exchange and encoding of identical information” amplifying over-squashing.

**Weaknesses:**

1. The necessity of k-redundant Neighborhood Tree (k-NT) seems week. In Table 2, experiments on EXP-class, the performance seems to be always higher when k is smaller. Removing all redundant nodes seems to be the best choice, why use k as an selection?
2. Experiments seems to be not sufficient enough to support the authors claim. For example in the abstract, authors claimed that the paper experimentally shows the method alleviates over-squashing. They have shown the results for synthetic datas in Table 2, but they are no experiments for real-world datasets to show this (such as experiments on long-range graph benchmark).
3. In the introduction section, the authors mentioned PathNNs and RFGNN as closely related works. Also in table 3, the authors highlighted the best results from polynomial time complexity in bold. However, it seems that they are no comparison with any methods having polynomial time complexity other than linear.

**Questions:**

1. For experiment results in Table 1, 3, authors highlighted the best results with polynomial time complexity methods, emphasizing that DAG-MLP has advantages in time. What is the time complexity of DAG-MLP in terms of big-O notation? Also, is there any inference time comparison for the inference time of each method (GIN, SPN, PathNN, DAG-MLP)?
2. Following weakness #4, is there are more baselines to compare with the paper method having a polynomial time complexity? What about the results of RFGNN mentioned for related works?
3. In Table 3, the performance IMDB-B and IMDB-M datasets are said to not applicable. However, in the Michel et al.$^{[1]}$, they do report the performance of PathNN-SP+(K=2) for datasets IMDB-D and IMDB-M. What do the authors mean by not applicable? Also, what path length K did the authors use for PathNN networks in Table 3?

[1] Michel et al., Path neural networks: Expressive and accurate graph neural networks, ICML 2023

---

> ### Author Response · Authors · 2023-11-16
>
> Thank you very much for your review.
>
> **W1** Since Theorem 1 applies only to $1$-NTs, it is currently conjectured that on graph level $0$-NTs are as expressive as $1$-NTs. However, since it has not been proven yet, the claim of being strictly more expressive than $1$-WL holds only for $1$-NTs. We show that it does not make sense to choose a very high $k$, because this introduces redundancy again, up until the point where the $k$-NTs are equal to the unfolding trees.
>
> **W2** Please refer to the global reply regarding further experiments and the LRGB.
>
> **W3** Please refer to the answer to **Q2** below.
>
> **Q1** Please refer to the global reply regarding the time complexity of DAG-MLP.
>
> **Q2** RFGNN does not have a polynomial but exponential time complexity. To the best of our knowledge, no methods currently exist that address the problem of reducing redundancy in graph neural networks and running in polynomial time.
>
> **Q3** As stated in our paper, for the comparison methods we used the results reported in [1]. We only reported the best results from this method and hence chose PathNN-P(K = 1) and PathNN-SP+(K = 3). We will add the parameter choice to the final version to clarify this. While results for PathNN-SP+(K=2) are reported, the performance is worse than that of PathNN-P(K = 1).
>
> **Presentation score**
> As you rated the presentation of our article with 1 but did not mention any points of criticism regarding the presentation in the review, we would like to ask for specific comments to help us improve the presentation of the manuscript.
>
> [1] Michel et al., Path neural networks: Expressive and accurate graph neural networks, ICML 2023

---

> > ### Comment · Reviewer_h7VD · 2023-11-21
> >
> > I thank the authors for their comprehensive response. I don't have any additional queries, and adjusted the presentation score.
> >
> > However, despite recognizing the constraints of limited time, I find it challenging to fully agree with the claim that DAG-MLP performs well in long-range tasks. The reported performance of GCN in LRGB$^{[1]}$ is approximately 0.593. I believe the authors' message would have been more clear, if they had compared one instance of DAG-MLP directly for the entire dataset, rather than testing many variants for sampled data.
> >
> > [1] Dwivedi et al., Long-range graph benchmark, 2022

---

### Official Review · Reviewer_e2WG · 2023-10-31

**Soundness:** 3 good
**Presentation:** 2 fair
**Contribution:** 2 fair
**Rating:** 5
**Confidence:** 3

**Summary:**

This paper proposes a new graph neural network architecture that alleviates the redundancy in the message-passing structure. The authors (1) prove that the expressive power of the new GNN architecture improves over the 1-WL test and (2) the new GNN architecture alleviates the over-squashing issue. The proposed architecture is evaluated on the synthetic datasets and the TUDataset.

**Strengths:**

This paper proposes a new GNN architecture to alleviate the redundancy of message passing. The proposed architecture is sound. The figures are helpful for understanding the paper.

**Weaknesses:**

My main concern is on the positioning of the paper with respect to similar work on alleviating GNN redundancy, i.e., RFGNN (Chen et al., 2022), and the weak experimental results.

### Comparison with RFGNN

- This work argues to alleviate over-squashing based on the results from RFGNN (Chen et al., 2022). However, as the authors argue, their proposed GNN architecture is different from RFGNN. Hence the logic is incomplete, i.e., it is not clear whether the proposed architecture alleviates over-squashing based on the same logic as RFGNN.
-  Upon reading Appendix A, the authors seem to claim that RFGNN introduces more redundancy compared to the proposed work. Since there is no clear explanation of how redundancy is harmful to GNN tasks, it is hard for me to understand the benefit of the proposed DAG-MLP.
- Furthermore, the authors do not compare the expressive power of DAG-MLP compared to RFGNN. One might argue that RFGNN might be more expressive than the proposed DAG-MLP at the cost of introducing more redundancy.
- In addition, the authors claim speed-up of RFGNN as another benefit. I wonder if the authors could empirically show this in a meaningful scenario, e.g., large-scale graphs.

### Weak experiments (TUDataset)
- Overall, I think TUDataset is not good enough for evaluating the performance of DAG-MLP in practical scenarios. Especially, to validate the ability of DAG-MLP to alleviate over-squashing, I strongly suggest the long-range graph benchmark (Dwivedi et al., 2022) to run the proposed DAG-MLP.
- The proposed work underperforms compared to the PathGNN. While the authors argue that PathGNN takes exponential running time, the actual running time is not reported. Hence it is hard to tell whether the issue is practically relevant.
- The statistical box plot in Appendix F should be similarly drawn for the baselines to make a fair comparison.
- The authors use four versions of DAG-MLP (0/1-NT, fixed single height/combined heights) while the relevant baselines have usually one or two versions (PathGNN has three versions, but DAG-MLP is not directly compared due to computational complexity). This makes the comparison unfair especially for TUDataset with high variance scores.
- The list of baselines is not comprehensive enough to check whether if performance improvement of the proposed DAG-MLP is practically relevant.

**Questions:**

How does the actual running time of DAG-MLP compare with the baselines in the considered experiments? I think this is an important criterion since the main (and possibly the only) benefit of DAG-MLP over RFGNN is the computational complexity.

---

> ### Author Response · Authors · 2023-11-16
>
> Thank you very much for your review and remarks, which we are happy to discuss.
>
> **Comparison to RFGNN**
>   * We elaborate on the relation to RFGNN regarding the ability to avoid over-squashing in the global reply. Our analysis also shows how redundancy is harmful to GNN tasks and that our approach provides advantages in this respect compared to MPNNs and RFGNN. We believe that these arguments lift your concerns.
>   * RFGNN claims to be fully expressive (solve graph isomorphism) and has exponential runtime. This is a huge drawback. Additionally, it does not address the second type of redundancy (computational redundancy), which we also address in our submission.
>   * Please refer to our global reply regarding time complexity.
>
> **Weak experiments**
>   * Please refer to the global reply regarding experiments on LRGB.
>   * Since we do not possess the same hardware as [1], comparing the running times of our method and PathNN is not feasible. We think that the theoretical time complexity is more meaningful, since it is independent of the hardware used and the optimization of the implementations.
>   * The statistical plot for the baselines can be found in Appendix 1, Figure 1 in [1]. In their plot, we only consider the test set (square symbol)
>   * We report the results for each of our variants separately, so a fair comparison is possible. It is often the case that only the best parameter choice of each model is shown and mixed throughout datasets. We did not do that to ensure a fair comparison.
>   * Please refer to our global reply regarding additional experiments.
>
> [1] Errica, F., Podda, M., Bacciu, D., & Micheli, A Fair Comparison of Graph Neural Networks for Graph Classification, ICLR 2020.
>
> **Questions**
> See answer to point 2 in “Weak experiments”. We also address this in our global reply. We will state the difference in computational complexity more clearly in the final version.

---

> ### Comment · Reviewer_e2WG · 2023-11-23
>
> Thank you for the response. I still find some of my concerns are not well resolved.
>
> I still think the actual running time should be reported. Empirical running time is important since it matters in practice. Since the PathNN has a public repository, the authors could simply reproduce the results and compare the runtime.
>
> I also find the newly reported results not very significant. There exist other baselines like DReW that perform better than the proposed method.

---

> > ### Author Response · Authors · 2023-11-23
> >
> > Thank you for considering our response. We agree that the empirical running time is of importance. However, reliable comparisons are also difficult to achieve, e.g., because implementations are optimized to a different extent.
> >
> > We are happy to consider DReW and other contemporaneous work in our comparison. However, we have to ensure that the same setup is used or reproduce the results for a fair comparison. For this reason, we compared to SJLR. However, our main focus is to show that our technique improves the message passing. We are not claiming to outperform rewiring techniques in general.

---

### Official Review · Reviewer_8Vsr · 2023-10-31

**Soundness:** 3 good
**Presentation:** 4 excellent
**Contribution:** 3 good
**Rating:** 6
**Confidence:** 3

**Summary:**

A compact representation of neighborhood trees is proposed, from which node and graph embeddings via a neural tree canonization technique are computed. The main goal is reduce redundancy in message passing GNNs to address oversquashing. The resulting message passing GNN is provably more expressive than the Weisfeiler-Leman test.

**Strengths:**

- Good literature discussion with details what distinguishes the different approaches.
- The basic concepts are introduced in detail.
- The proposed 1-NT isomorphism test is provably more powerful than the Weifeiler-Leman test.
- Experiments verify that a reduction in redundancy could help address oversquashing.

**Weaknesses:**

- k seems to be a hyper-parameter that would need to be tuned in practice.
- Even though DAG-MLPs are provably more expressive than the Weisfeller-Lehmann method, they are not proven to be fully expressive (i.e. distinguish any non-isomorphic graphs).
- The computational complexity of the proposed architecture and algorithms are not analysed but form an integral part of the contribution.
- PathNN-P seems stronger on the Enzymes and Proteins dataset but also suffers from exponential computational time complexity.

**Questions:**

- How expressive are the proposed DAG-MLPs? It seems like there could exist non-isomorphic graphs that cannot be distinguished by DAG-MLP. What would be an example?
- How does the expressive power compare with baseline methods?
- What is the computational complexity of building and evaluation DAG-MLPs? What are their memory requirements?
-> It would be helpful to add measurements of time complexity in the tables of the experiments.
- What could be other explanations why k-NTs perform less well for higher $k$ in Table 3? Does the explanation have to be over-squashing?

---

> ### Author Response · Authors · 2023-11-16
>
> Thank you very much for the review, the remarks and the questions, which we are happy to discuss. Since the weaknesses and questions overlap, we discuss them jointly if possible. Please let us know, if anything is not adequately addressed.
>
> **Weaknesses and Questions**
>   * $k$ as a hyper-parameter: Setting $k$ to $1$ (or even $0$) should in general be the best solution. With higher $k$ values, redundancy is introduced again, up until the point where the $k$-NTs are equal to the unfolding trees. We do not expect NTs with $k$ greater than $1$ to be more expressive than $1$-NTs, therefore this parameter does not have to be optimized (other than maybe choosing between $0$ and $1$). If we could prove that $0$-NTs are as expressive on graph-level as $1$-NTs, then $0$-NTs could be used in any case for graph-level tasks.
>   * Not fully expressive/Example: Indeed, DAG-MLP is not fully expressive. There are non-isomorphic graphs that it cannot distinguish (for example the $4\times 4$-Rooks and Shrikhande graph). There is always a trade-off between running time and expressivity. Currently, no polynomial time algorithms for solving graph isomorphism are known.
>   * We analyzed the size of the DAG, as well as the time to build it in our submission in Section 4.2. This makes up the crucial part of the running time for our method.  Please also refer to the global reply regarding the computational complexity.
>   * Please refer to the global reply regarding the expressivity of the baseline methods.
>   * We believe that over-squashing is a suitable explanation for our experimental findings. We have formally related our method to over-squashing in the global reply. Using the notation introduced there, we have $I_{i\text{-NT}}(v,u) \leq I_{j\text{-NT}}(v,u)$ for $i>j$ supporting this hypothesis.

---

> > ### Comment · Reviewer_8Vsr · 2023-11-22
> >
> > I thank the authors for their response and do not have any further open questions.
> > As my main concerns have been addressed, I will keep my score, but will be open to adapting it during the discussion with other reviewers.

---

### Official Review · Reviewer_zntd · 2023-11-01

**Soundness:** 3 good
**Presentation:** 3 good
**Contribution:** 2 fair
**Rating:** 3
**Confidence:** 3

**Summary:**

This paper proposes a novel aggregation scheme based on neighborhood trees to control redundancy in message-passing graph neural networks (MPNNs). The authors show that reducing redundancy improves expressivity and experimentally show that it alleviates over squashing.

**Strengths:**

1) The paper introduces a novel aggregation scheme based on neighborhood trees, which allows for controlling redundancy in message passing MPNNs.
2) The authors provide a theoretical analysis of expressivity that shows the proposed method is more expressive than the Weisfeiler-Leman method.

**Weaknesses:**

1) The main weakness is the computational cost, which requires O(nm) space where n is the number of nodes and m is the number of edges. This brings a significant limitation to the applicability of the proposed method, even for moderate-sized graphs.

2) The experimental result only shows occasional marginal improvements over some baselines and only on a few datasets. This is not enough to demonstrate the effectiveness of the proposed method.

3) One main motivation for the proposed method is to address over squashing, but there is no theoretical analysis of the proposed method to address it.

**Questions:**

1) What is the largetst graph size that the proposed method can handle?
2) What is the preprocessing time for the proposed method?

---

> ### Author Response · Authors · 2023-11-16
>
> Thank you very much for the review. We address some of your remarks in our global reply to all reviewers and look forward to discussing further comments.
>
> **W1** We agree that the increase in running time is a drawback of our approach. However, our method has significantly lower computational complexity than comparable approaches. We elaborate on this further in the global reply.
>
> **W2** We compare our approach to the most recent state-of-the-art methods in Table 3. Since these methods have been shown to outperform the baselines, we believe that this is appropriate. In [1] there are results for other methods, but Path-NN outperforms them in most cases. We provide additional experimental results supporting the advantages of our method in the global reply.
>
> **W3** Thank you for asking how our approach formally relates to over-squashing. We referred to the analysis in [2] without making the relation explicit in our manuscript. We have elaborated this in the global reply showing that our approach can formally be related to over-squashing and improves in this respect over existing methods. We believe that including these arguments will strengthen our manuscript.
>
> **Q1** This depends on the properties of the graphs and on the hardware used. Giving a general answer to this question is generally difficult (also for standard GNNs).
>
> **Q2** In practice, this again depends on the hardware used. On a 64 CPU our non-optimized implementation performs the preprocessing for the 600 protein graphs of the dataset ENZYMES (33 nodes and 62 on an average) in 7.8 seconds for 1-NTs. We believe that the theoretical time complexity is more meaningful (although it does not take our compression techniques into account) and refer to the global reply.
>
> [1] Michel et al., Path neural networks: Expressive and accurate graph neural networks, ICML 2023
>
> [2] Rongqin Chen, Shenghui Zhang, Leong Hou U, Ye Li: Redundancy-Free Message Passing for Graph Neural Networks. NeurIPS 2022

---

### Author Response · Authors · 2023-11-16

We would like to thank the reviewers for their valuable feedback. We address issues raised by multiple reviewers here and provide individual answers to the reviewers below.
Following the suggestions, we provide additional experimental results, clarify the relation to over-squashing and RFGNN, and state the computational complexity more clearly both here and in the final version of our submission.

---

> ### Author Response · Authors · 2023-11-16
> **Additional Experimental Results**
>
> Several reviews requested additional experiments. We performed further experiments on two different types of tasks and datasets, showing clear advatages of our method for specific datasets.
>
>
> ## Node Classification under Heterophily and Homophily
>
> We provide additional results for node classification datasets on both homophily and heterophily datasets. These datasets differ regarding their homophily ratio, i.e., the fraction of edges in a graph which connect nodes that have the same class label [1]. Heterophily tasks are particularly challenging for standard GNNs [1] as they require capturing the structure of neighborhoods instead of “averaging” over the neighboring features. We provide results from the paper [2] including the state-of-the-art graph rewiring technique SJLR combined with SGC and GCN, which performs best in the evaluation. We performed experiments with GIN and DAG-MLP using the same data splits as [2]. We report the best results for $l$ layers with $l \in \\{2,3,4\\}$ and four different combine methods for GIN and DAG-MLP.
>
> | Model                            | Texas |  Wisconsin | Cornell | Cora |  CiteSeer | PubMed |
> |----------------------------------|-------|------------|---------|--------|----------|--------|
> | Homophily ratio                  | 0.11  | 0.21       | 0.3     | 0.8  | 0.74     | 0.8    |
> |                                  |       |            |         |             |          |        |
> | GCN |  58.05 ± 0.9 | 52.10 ± 0.9 | 67.34 ± 1.5 | 81.81 ± 0.2 |  68.35 ± 0.3 | 78.25 ± 0.3 |
> | SJLR-GCN | 60.13±0.89 | 55.16±0.95 | **71.75**±1.50 | **81.95**±0.25 | **69.50**±0.33 | **78.60**±0.33 |
> | SGC | 56.69 ± 1.7 | 47.90 ± 1.7 |  53.40 ± 2.1 | 76.9 ± 1.3 |  67.45 ± 0.8 | 71.79 ± 2.1 |
> |SJLR-SGC | 58.40±1.48 | 55.42±0.92 | 67.37±1.64 |  81.24±0.77 | 68.39±0.69 | 76.28±0.96 |
> |                                  |       |            |         |             |          |        |
> | GIN                          | 75.68 ± 6.4 | 77.84 ± 6.5 | 68.92 ± 5.3 | 76.76 ± 1.4 |  64.49 ± 1.5 | 76.46 ± 1.1 |
> |                                  |       |            |         |             |          |        |
> | DAG-MLP (k=0)   | 79.19 ± 8.2 | 78.82 ± 4.1 | 71.08 ± 5.4 | 74.01 ± 2.0 |  60.55 ± 3.6  | 75.33 ± 1.1 |
> | DAG-MLP (k=1)    | **79.46** ± 8.1 | **80.78** ± 4.5 | 70.81 ± 4.6 | 74.54 ± 1.4 |  61.09 ± 1.5 | 75.53 ± 1.1 |
>
>
> DAG-MLP outperforms GIN on the heterophily datasets (those with low homophily ratio), while GIN is better on homophily ones. The results indicate that our neighborhood trees are able to capture the relevant neighborhood structure more accurately than unfolding trees used by GIN. Our method outperforms SJLR on the two heterophily datasets Texas and Wisconsin.
>
> [1] Zhu et al., Beyond Homophily in Graph Neural Networks: Current Limitations and Effective Designs, NeurIPS 2020.
>
> [2] Giraldo, Jhony H. et al. “On the Trade-off between Over-smoothing and Over-squashing in Deep Graph Neural Networks.” ACM International Conference on Information and Knowledge Management 2022.
>
> ## Long Range Graph Benchmark (LRGB)
>
> We performed experiments on the two LRGB datasets Peptides-Functional and Peptides-Structural comparing GCN, GIN and DAG-MLP using the same experimental setup. Due to the limited time available, we report the results on a subset of 1553 graphs randomly sampled from the full datasets. We used  $l$ layers with $l \in \\{0,1,\dots,8\\}$ for all methods.
>
> |         Model         | AP ↑ (Peptides Functional) | MAE ↓ (Peptides Structural) |
> |-----------------------|-------------------|-------------------|
> | GCN - NONE Combine    | 0.310 ± 0.018      | 0.671 ± 0.002     |
> | GCN - SUM Combine     | 0.309 ± 0.006      | 0.664 ± 0.006     |
> | GCN - MEAN Combine    | 0.284 ± 0.025      | 0.641 ± 0.016     |
> | GCN - CAT Combine     | 0.336 ± 0.012      | 0.614 ± 0.010     |
> | GIN - NONE Combine    | 0.359 ± 0.013      | 0.562 ± 0.038     |
> | GIN - SUM Combine     | 0.408 ± 0.017      | 0.492 ± 0.021     |
> | GIN - MEAN Combine    | 0.408 ± 0.010      | 0.489 ± 0.019     |
> | GIN - CAT Combine     | 0.433 ± 0.000      | 0.460 ± 0.004     |
> | DAG-MLP - NONE Combine | **0.488 ± 0.023**      | 0.584 ± 0.033     |
> | DAG-MLP - SUM Combine  | 0.474 ± 0.017      | 0.455 ± 0.001     |
> | DAG-MLP - MEAN Combine | 0.467 ± 0.006      | **0.450 ± 0.007**     |
> | DAG-MLP - CAT Combine  | 0.472 ± 0.005      | 0.525 ± 0.026     |
>
> Our preliminary results show that DAG-MLP performs well in long-range tasks, supporting our theoretical findings that our method is less susceptible to over-squashing. We will add experimental results for the full datasets as soon as possible.

---

> ### Author Response · Authors · 2023-11-16
> **Computational Complexity and Expressivity**
>
> Since practical computation time is very dependent on implementation details, as well as the hardware used, a fair comparison of methods in terms of running time is difficult. The theoretical complexity of our method is stated in Section 4.2, however, we will present it more clearly in a final version.
> The DAG representing the $k$-NT of a single node has a size in $O(m \cdot (k+1)),$ where $m$ is the number of edges in the graph. The DAG can also be generated in this time.
> The lexicographic encoding and merging can then be done in time linear to the DAGs size.
> A trivial upper bound on the size of the merge DAG of a graph with $n$ nodes and $m$ edges is $O(nmk + nm)$. Overall, this means that preprocessing can be done in $O(nmk)$ time, where $k$ can be considered constant. For $1$-NTs, which have been proven to be more expressive than $1$-WL, we obtain time $O(nm)$. The following table compares the complexity and expressivity of the different methods.
>
> |Method | Complexity of Preprocessing | Size of Computation Graph/Running time | Expressivity|
> |---- | ---- | ----- | ----|
> |DAG-MLP (k=1) | $O(nm)$  | $O(nm)$ | $>1$-WL|
> |PathNet | $O(mb)$ |$O(2^L(m+m_2)$|na|
> |PathNN-SP | $O(nb^K)$| $O(nbk)$ | incomparable|
> |PathNN-SP+ | $O(nb^K)$| $O(nbk)$ |$>1$-WL|
> |RFGNN | $O\left(\\frac{n!}{(n-h-1)!}\right)$|$O\left(\\frac{n!}{(n-h-1)!}\right)$|$>1$-WL|
>
>   * $n$: number of nodes
>   * $m$: number of edges
>   * $b$: maximum node degree
>   * $K$: path length
>   * $L$: number of layers
>   * $m_2 = 0.5 \sum{v \in V} \mid N_2(v)\mid$
>
> Relating the expressivity of the different approaches is non-trivial. For PathNet the expressivity in terms of the WL-hierarchy is not investigated. Except for PathNN-SP, all other approaches are (on graph level) more expressive than $1$-WL and RFGNN claims to be maximally expressive. PathNN-SP "can only disambiguate graphs at least as well as the WL algorithm and are [/is] not strictly more powerful, since also isomorphic graphs could be mapped to different representations." [1] It is unclear whether DAG-MLP (k=1) and PathNN-SP+ have the same expressivity, whether one is strictly more expressive than the other, or whether they are incomparable.
>
> In RFGNN all possible paths and in PathNN-SP+ all shortest paths are enumerated. As can be seen in a very simple example of a graph consisting only of a chain of joined 4-cycles, there is an exponential number of such paths, which still contain redundancy. In our architecture we avoid this redundancy by not building the NTs explicitly, but generating DAGs, which are much more compact.
>
> Summarizing the results, our method is the first to address redundancy in GNNs with polynomial running time. We believe that our method fills a gap not yet considered in GNN research.
>
> [1] Michel et al., Path neural networks: Expressive and accurate graph neural networks, ICML 2023

---

> ### Author Response · Authors · 2023-11-16
> **Theoretical Analysis of Over-Squashing and Comparison with RFGNN**
>
> Reviewer zntd criticized that the relation to over-squashing is not theoretically analyzed. Indeed, we only referred to the analysis provided in the paper [1] introducing RFGNN. Reviewer e2WG argues that the advantages over RFGNN [1] are not clear. We would like to address both concerns jointly.
>
> Several authors [1, 2, 3, 4] developed and refined techniques to measure the influence of a node $v$ with initial node feature $\mathbf{x}_v$ on the output $\mathbf{h}_u^{(k)}$ of a node $u$ after layer $k$ by the Jacobian as
> $\\partial \\mathbf{h}_u^{(k)}/ \\partial \\mathbf{x}_v$.
>
> Following [1, Lemma 3] we obtain (under the assumptions commonly used) that the relative influence of $v$ on $u$ is
>
> $$
> I(v,u)=
> \mathbb{E}\left(\frac{\partial \mathbf{h}\_{u}^{(k)}/ \partial \mathbf{x}_{v}}{\sum\_{w\in V} \partial \mathbf{h}\_{u}^{(k)}/ \partial \mathbf{x}\_{w}}\right)=
> \frac{[\hat{A}^k]\_{u,v}}{\sum\_{w\in V}[\hat{A}^k]\_{u,w}},
> $$
>
> where $\hat{A}=A+I$ is the adjacency matrix of the graph $G$ with added self-loops. Note that $[\hat{A}^k]_{u,v}$ is the number of walks from $u$ to $v$ (and vice versa) in $G$ of length at most $k$.
> Over-squashing occurs if $I(v,u)$ becomes too small, i.e., only a small fraction of walks of length up to $k$ ending at $u$ start at $v$.
>
> This idea can easily be linked to the concept of unfolding trees underlying our work. Consider the unfolding tree $F^u_k$ of the node $u$ with height $k$. It follows from its construction that there is a bijection between walks of length at most $k$ ending at $u$ in $G$ and paths in $F^u_k$ from some node to the root (see [5] for details on unfolding trees and walks). Hence, pruning the unfolding tree has an effect on walk counts and, thus, on the relative influence. Consider the example in Figure 1 in our paper and let $u$ be the red node and $v$ the yellow node. For unfolding trees we obtain a relative influence of $I_{\text{MPNN}}(v,u)= \frac{1}{8}$, and for $0$-NTs a relative influence of $I_{0\text{NT}}(v,u)= \frac{1}{4}$, showing that NTs have the potential to reduce over-squashing.
>
> We can also formally show in this framework that our method is less susceptible to over-squashing than MPNNs and RFGNN [1]. Consider a node $v$ and a node $u$ with shortest-path distance $k$. To pass information from $v$ to $u$ we require (at least) $k$ layers. We consider the unfolding tree (MPNN), the $0$- and $1$-NT (our approach) and the TPT (RFGNN), all of height $k$. In all trees the node $v$ occurs in the last level only, i.e., as a leaf of the tree. The number of occurrences of $v$ is equal in all trees, since all walks and simple paths of length $k$ reaching $v$ are shortest paths. Hence, the numerator of the relative influence is equal for all methods. However, $0$- and $1$-NTs are subtrees of unfolding trees (according to their definition). Moreover, $0$-NTs are subtrees of TPTs, since they contain shortest paths only instead of all simple paths (and cycles). Also, $1$-NTs are subtrees of TPTs, since they contain simple paths only, but not all of them. Hence, we can relate the total number of nodes in the trees, i.e., walks contributing to the denominator of the relevant information, and obtain
> $$I_{\text{MPNN}}(v,u)\leq I_{\text{TPT}}(v,u) \leq I_{1\text{NT}}(v,u) \leq I_{0\text{NT}}(v,u).$$
> This theoretically shows that our method provides advantages regarding over-squashing using the formalization developed in recent papers. Moreover, it establishes a relationship between our approach and RFGNN, showing that our method alleviates the over-squashing problem more effectively. We will incorporate these arguments in our manuscript.
>
>
>
> [1] Rongqin Chen, Shenghui Zhang, Leong Hou U, Ye Li: Redundancy-Free Message Passing for Graph Neural Networks. NeurIPS 2022
>
> [2] Keyulu Xu, Chengtao Li, Yonglong Tian, Tomohiro Sonobe, Ken-ichi Kawarabayashi, Stefanie Jegelka: Representation Learning on Graphs with Jumping Knowledge Networks. ICML 2018: 5449-5458
>
> [3] Jake Topping, Francesco Di Giovanni, Benjamin Paul Chamberlain, Xiaowen Dong, Michael M. Bronstein: Understanding over-squashing and bottlenecks on graphs via curvature. ICLR 2022
>
> [4] Francesco Di Giovanni, Lorenzo Giusti, Federico Barbero, Giulia Luise, Pietro Lio, Michael M. Bronstein: On Over-Squashing in Message Passing Neural Networks: The Impact of Width, Depth, and Topology. ICML 2023: 7865-7885
>
> [5] Nils M. Kriege: Weisfeiler and Leman Go Walking: Random Walk Kernels Revisited. NeurIPS 2022

---

### Meta-Review · Area_Chair_XCdd · 2023-12-12

**Metareview:**

Summary: The article proposes an aggregation scheme to control redundancy in message passing graph neural networks, show this increases expressivity and helps reduce over squashing.

Strengths: Novel aggregation scheme with expressivity guarantees.

Weaknesses: A main concern brought up in the reviews is the computational cost. The authors agree this is a drawback. The concern persisted after the rebuttal. Further concerns were that experiments showing only occasional improvements or missing baselines, and the positioning of the paper with respect to other works. Reviewers have provide suggestions.

The article takes an interesting path to addressing relevant problems. However, at the moment the shortcomings outweigh the strengths. Hence I must reject this submission.

**Justification For Why Not Higher Score:**

Parts of the objectives on oversquashing are not sufficiently supported and there are reservations with computational cost and improvements over baselines.

**Justification For Why Not Lower Score:**

NA

---

### Decision · Program_Chairs · 2024-01-16

Reject